# A quantitative approach for analyzing the spatio-temporal distribution of 3D intracellular events in fluorescence microscopy

Thierry Pécot[1], Liu Zengzhen[2], Jérôme Boulanger[2], Jean Salamero[2,3], Charles Kervrann[1]*

[1]Serpico Team-Project, Inria, Centre Rennes-Bretagne Atlantique, Rennes, France; [2]CNRS UMR 144, Space Time Imaging of Endomembranes Dynamics Team, PSL Research University, Institut Curie, Paris, France; [3]Cell and Tissue Imaging Facility, IBiSA, Institut Curie, Paris, France

**Abstract** Analysis of the spatial distribution of endomembrane trafficking is fundamental to understand the mechanisms controlling cellular dynamics, cell homeostasy, and cell interaction with its external environment in normal and pathological situations. We present a semi-parametric framework to quantitatively analyze and visualize the spatio-temporal distribution of intracellular events from different conditions. From the spatial coordinates of intracellular features such as segmented subcellular structures or vesicle trajectories, QuantEv automatically estimates weighted densities that are easy to interpret and performs a comprehensive statistical analysis from distribution distances. We apply this approach to study the spatio-temporal distribution of moving Rab6 fluorescently labeled membranes with respect to their direction of movement in crossbow- and disk-shaped cells. We also investigate the position of the generating hub of Rab11-positive membranes and the effect of actin disruption on Rab11 trafficking in coordination with cell shape.
DOI: https://doi.org/10.7554/eLife.32311.001

*For correspondence:
Charles.Kervrann@inria.fr

**Competing interests:** The authors declare that no competing interests exist.

## Introduction

Modern light microscopy associated with fluorescence molecule tagging allows studying the spatial distribution of intracellular events. Unfortunately, fluorescent images are complex to analyze and additional software is needed to evaluate statistical differences between different conditions (*Meijering et al., 2016*; *Tinevez et al., 2017*). Automatic methods have the obvious advantage of being quicker and reproducible. However, most computational methods are based on the complex combination of heterogeneous features such as statistical, geometrical, morphological and frequency properties (*Peng, 2008*), which makes it difficult to draw definitive biological conclusions. Additionally, most experimental designs, especially at single-cell level, pool together data coming from replicated experiments of a given condition (*Schauer et al., 2010*; *Merouane et al., 2015*; *Biot et al., 2016*), neglecting the biological variability between individual cells.

Micro-patterning is now a well-established strategy to reduce morphological variability by imposing constraints on adhesion sites, which has been shown to influence the cytoskeleton geometry and transport carrier localization (*Théry et al., 2005*; *Schauer et al., 2010*). This technique opened the way to pairwise comparisons of conditions with a two-sample kernel density-based test by pooling together all data from each condition (*Duong et al., 2012*). Unfortunately, it does not consider the sample-to-sample variability because all replicated experiments from a given condition are simply merged together. Additionally, the visualization of the kernel density maps enables to average

**eLife digest** Proteins are the workhorses of the body, performing a range of roles that are essential for life. Often, this requires these molecules to move from one location to another inside a cell. Scientists are interested in following an individual protein in a living cell 'in real time', as this helps understand what this protein does.

Scientists can track the whereabouts of a protein by 'tagging' it with a fluorescent molecule that emits light which can be picked up by a powerful microscope. This process is repeated many times on different samples. Finally, researchers have to analyze all the resulting images, and conduct statistical analysis to draw robust conclusions about the overall trajectories of the proteins. This process often relies on experts assessing the images, and it is therefore time-consuming and not easily scalable or applied to other experiments.

To help with this, Pécot et al. have developed QuantEV, a free algorithm that can analyze proteins' paths within a cell, and then return statistical graphs and 3D visualizations. The program also gives access to the statistical procedure that was used, which means that different experiments can be compared.

Pécot et al. used the method to follow the Rab6 protein in cells of different shapes, and found that the conformation of the cell influences where Rab6 is located. For example, in crossbow-shaped cells, Rab6 is found more often toward the three tips of the crossbow, while its distribution is uniform in cells that look like disks. Another experiment examined where the protein Rab11 is normally placed, and how this changes when the cell's skeleton is artificially disrupted. Both studies help to gain an insight into the behavior of the cellular structures in which Rab6 and Rab11 are embedded.

Following proteins in the cell is an increasingly popular method, and there is therefore a growing amount of data to process. QuantEV should make it easier for biologists to analyze their results, which could help them to have a better grasp on how cells work in various circumstances.

DOI: https://doi.org/10.7554/eLife.32311.002

several experiments but fails to identify specific locations of interest in the cell (e.g. docking areas). Finally, assessing the dynamical behavior of labeled membrane structures, a fundamental task for trafficking analysis, remains out of scope in this framework.

In this paper, we describe a method that we call QuantEv dedicated to the analysis of the spatial distribution of intracellular events represented by any static or dynamical descriptor (e.g. detected points, segmented regions, trajectories...) provided that the descriptors are associated with spatial coordinates. QuantEv offers a unifying frame to decipher complex trafficking experiments at the scale of the whole cell. It is typically able to detect subtle global molecular mechanisms when trajectory clustering fails. An overview of the approach is presented in *Figure 1*. Our approach first computes 3D histograms of descriptors in a cylindrical coordinate system (parameterized by radius $r$, angle $\theta$ and depth $z$) with computational cell shape normalization, enabling comparisons between cells of different shape. Densities are obtained via adaptive kernel density estimation (*Silverman, 1986*; *Taylor, 2008*). Visualization through histograms and densities allows giving a clear biological interpretation of the experiments. We use the Earth Mover's Distance (*Rubner et al., 2000*) and the Circular Earth Mover's Distance (*Rabin et al., 2011*) to measure the dissimilarity between densities associated with different experimental conditions. A statistical analysis of these distances reliably takes into account the biological variability over replicated experiments. By computing weighted densities for each point in the cell as the reference center, QuantEv identifies the point that gives the most uniform angular distribution. This point may coincide with a biological structure of interest that would act as the events emitter or attractor.

In the section Results, we describe the application of QuantEv to detect significant differences between molecular trafficking and phenotypes observed in cells with various shapes. The first application is concerned with the distribution of membranes labeled by GFP-Rab6 as a hallmark of vesicular carriers in unconstrained, crossbow- and disk-shaped cells. Rab6 proteins are transiently anchored to moving transport carriers from the Golgi apparatus located at the cell center to Endoplasmic Reticulum entry sites or to plasma membrane (*White et al., 1999*; *Chavrier and Goud,*

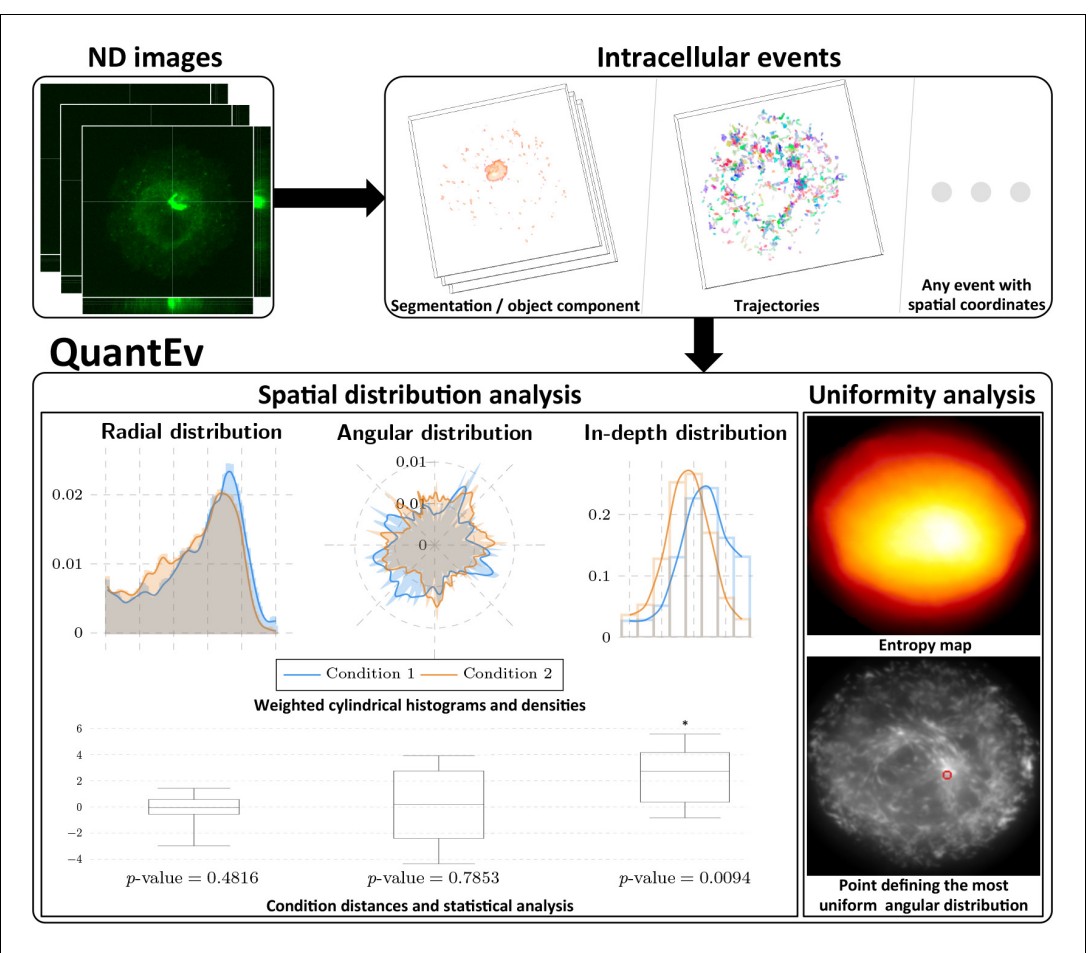

**Figure 1.** Overview of QuantEv approach. Spatial distribution analysis: QuantEv computes 3D histograms and densities of intracellular descriptors to quantitatively compare different experimental conditions. Uniformity analysis: QuantEv identifies the point that gives the most uniform angular distribution of intracellular descriptors. DOI: https://doi.org/10.7554/eLife.32311.003

*1999*; *Echard et al., 2000*; *Opdam et al., 2000*; *Grigoriev et al., 2007*; *Bardin et al., 2015*), both assumed to be located at the cell periphery. Cell shape imposes constraints on the cytoskeleton and consequently influences the spatial distribution of Rab6 transport carriers, as confirmed with kernel density maps (*Schauer et al., 2010*). We apply QuantEv to visualize and quantify this influence and to localize regions in the cell associated with Rab6 trafficking stages. In addition, Rab6 positive membranes were reported to move from and toward the Golgi in apparent close proportions (*Grigoriev et al., 2007*, *Grigoriev et al., 2011*), and yet these membrane associated proteins are believed to traffic in majority from the Golgi located at the cell center to the cell periphery (*White et al., 1999*; *Chavrier and Goud, 1999*; *Echard et al., 2000*; *Opdam et al., 2000*; *Grigoriev et al., 2007*) where they should dissociate from membranes and recycle back to the cytosol. To investigate these apparently antagonist statements, we apply QuantEv on Rab6 trajectories to characterize the dynamical behaviors of these transport carriers.

The second application focuses on the dynamics of mCherry-Rab11-positive membranes. Rab11 is known to be primarily localized to the Endosomal Recycling Compartment (ERC), and it organizes spatially and temporally recycling from this compartment (*Ullrich et al., 1996*; *Gidon et al., 2012*; *Baetz and Goldenring, 2013*; *Boulanger et al., 2014*). Here, we confirm by using QuantEv the hypothesis that the labeled transport intermediates are uniformly distributed around the ERC at the plasma membrane plane. Furthermore, we also investigate the progressive effect of actin disruption induced by Latrunculin A injection on the ERC localization with respect to time. We finally apply

QuantEv to analyze the joined influence of actin disruption and cell shape on the radial distribution of Rab11 vesicles trafficking.

## Results

### Visualizing and quantifying the influence of cell shape on the spatial distribution of Rab6 positive membranes

We applied the QuantEv approach to visualize the spatial distribution of Rab6-positive membranes in unconstrained, crossbow- and disk-shaped cells (see *Appendix 1—figure 1*) and quantify their differences. To test the generic performance of QuantEv, these image sequences were acquired with two different 3D imaging modalities, a multi-point confocal microscopy and a wide field video microscopy. We compared the results obtained with QuantEv to those obtained with the more conventional kernel density (KD) maps (*Schauer et al., 2010*; *Merouane et al., 2015*). The KD approach concludes that the distribution of Rab6-positive membranes are clearly different between cells of different shapes (see *Figure 2a–c*, p value = 0 when considering unconstrained cells versus crossbow-shaped cells, unconstrained cells versus disk-shaped cells, and crossbow-shaped cells versus disk-shaped cells). Unfortunately, it also leads to a significant difference when image sequences with the same cell shape are compared (see *Figure 2d*). This demonstrates that the KD approach is too sensitive. Instead, QuantEv shows a uniform range of p values when cells with same shape are compared (see *Figure 2d*) while it leads to significant differences between radial, angular and in-depth distributions of Rab6 proteins from cells with different cell shapes (see *Figure 2e–f*). The angular distribution of Rab6 proteins is different for the three cell shapes. It ranges from a completely uniform distribution for disk-shaped cells, to a less regular distribution for unconstrained cells and to a distribution oriented toward the three tips of the crossbow for crossbow-shaped cells. In-depth and radial distributions are similar for crossbow- and disk-shaped cells. In contrast, they are different from unconstrained cells. Unconstrained cells show diverse sizes with a strong tendency to spread. This explains why the in-depth distribution is flatter for the unconstrained cells than for the constrained cells. Interestingly, QuantEv is able to reflect these differences. QuantEv also highlights a distribution maximum for a radius at the two-thirds (resp. five-sixth) the distance between the Golgi region border and the cell periphery for both micro-patterns (resp. unconstrained cells) (see *Figure 2f–i*). These maxima correspond to an accumulation of Rab6-positive membranes and identify the area where they enter a docking phase before switching to a tethering phase. The localization difference for these maxima between constrained and unconstrained cells is explained by a smaller adhesion area without micro-patterns, pushing the docking phase for vesicles closer to the cell periphery. Both radial and angular distributions unraveled by QuantEv represent a measurement of the environment constraints undergone by living cells.

### Inwards and outwards Rab6-positive membranes show two distinctive dynamical behaviors

Rab6-positive membranes are trafficking from the Golgi located at the cell center to the cell periphery (*White et al., 1999*; *Chavrier and Goud, 1999*; *Echard et al., 2000*; *Opdam et al., 2000*; *Grigoriev et al., 2007*) and at the same time move from and toward the Golgi in comparable proportions (*Grigoriev et al., 2007*; *Grigoriev et al., 2011*). To reconcile these two antagonist statements, we applied QuantEv as follows. Rab6 trajectories were classified into two categories (*Figure 3a–c*): (i) vesicles moving toward the cell periphery; (ii) vesicles moving toward the Golgi. As shown in *Figure 3d–f*, the proportion of Rab6-positive membranes moving toward the cell periphery and toward the Golgi are close (0.531 versus 0.469 for unconstrained cells, 0.497 versus 0.503 for crossbow-shaped cells, 0.521 versus 0.479 for disk-shaped cells). However, the radial distributions shown in *Figure 3d–f* display two distinctive modes for vesicles moving toward the cell periphery and those moving toward the Golgi (p value = 0.0002 for unconstrained cells, p value = 0.021 for crossbow-shaped cells, p value = 0.0008 for disk-shaped cells). Between the Golgi and the distribution maxima shown in *Figure 2f*, Rab6 vesicles are predominantly moving toward the cell periphery. Between these maxima and the cell periphery, they are in majority moving toward the Golgi, indicating that during their docking-tethering phase, the vesicles are predominantly moving toward the cell center. These two distinctive dynamical behaviors are consistent with the aforementioned antagonist

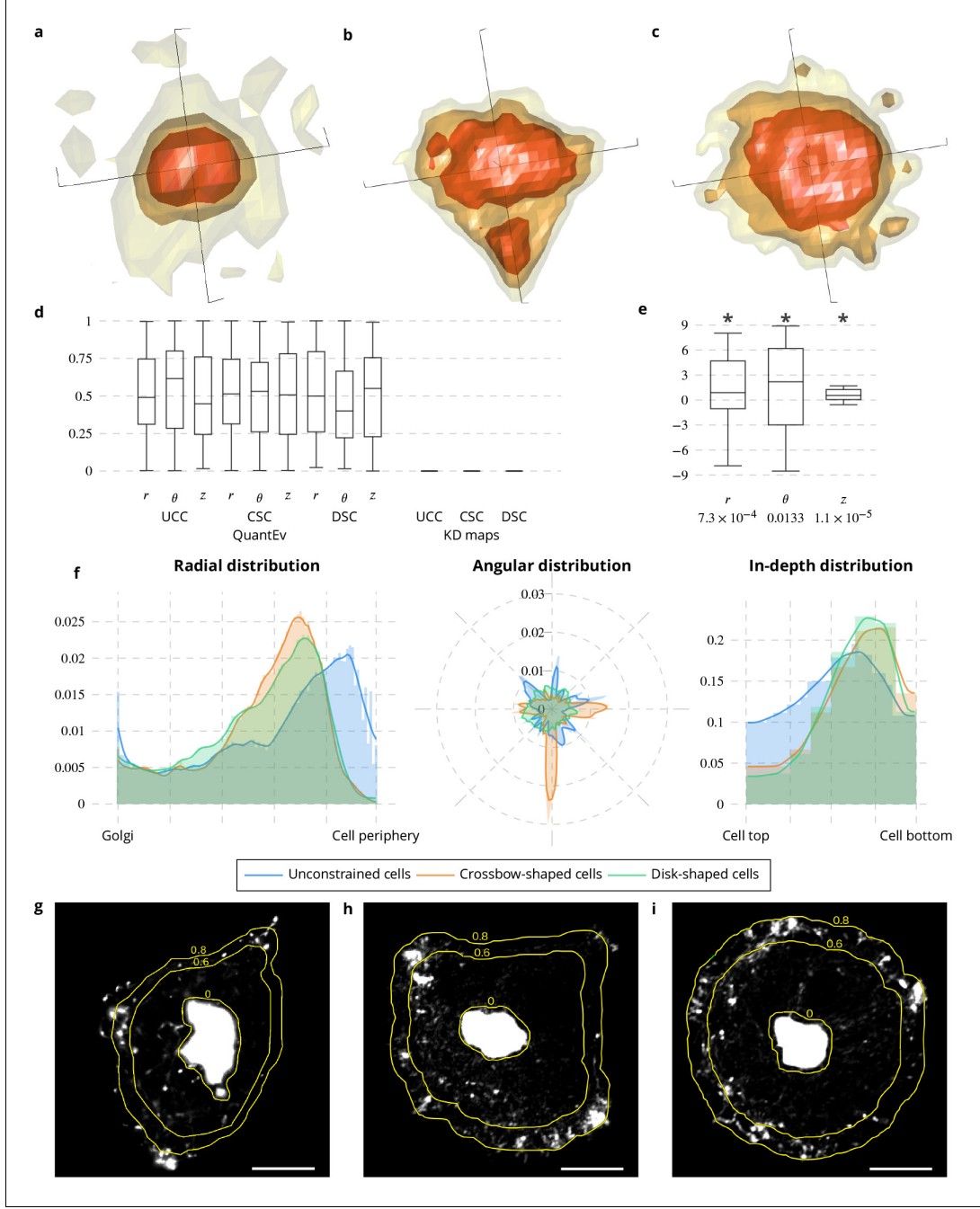

**Figure 2.** Spatial distribution analysis of Rab6 proteins. (a−c) 3D KD maps obtained with kernel density maps when considering all image sequences with unconstrained (**a**), crossbow- (**b**) and disk- (**c**) shaped cells. (**d**) Box and whisker plots of the p values obtained when comparing randomly 100 times two groups of unconstrained (UCC), crossbow-shaped (CSC) or disk-shaped cells (DSC) with QuantEv and KD maps. (**e**) Box and whisker plots of the condition differences with respect to radius $r$, angle $\theta$ and depth $z$ over 58 image sequences. p values under conditions of one-sided Wilcoxon signed-rank test when considering the condition differences are indicated below the box and whisker plots. A star (*) indicates that the p value is smaller than 0.05. (**f**) Histograms (bar plots) and densities (lines) of the spatial distribution of Rab6 positive membranes with respect to radius $r$, angle $\theta$ and depth $z$. These distributions come from 18 image sequences with an unconstrained cell (blue bar plots and lines), 18 image sequences with a crossbow-shaped cell (orange bar plots and lines) and 22 image sequences with a disk-shaped cell (green bar plots and lines). (**g−i**) Overlay of the average intensity projection map of an image sequence with an unconstrained (**g**), crossbow- (**h**) and disk- (**i**) shaped cell and the radial levels at 0.6 and 0.8. The scale bars correspond to 5 $\mu$m.

DOI: https://doi.org/10.7554/eLife.32311.004

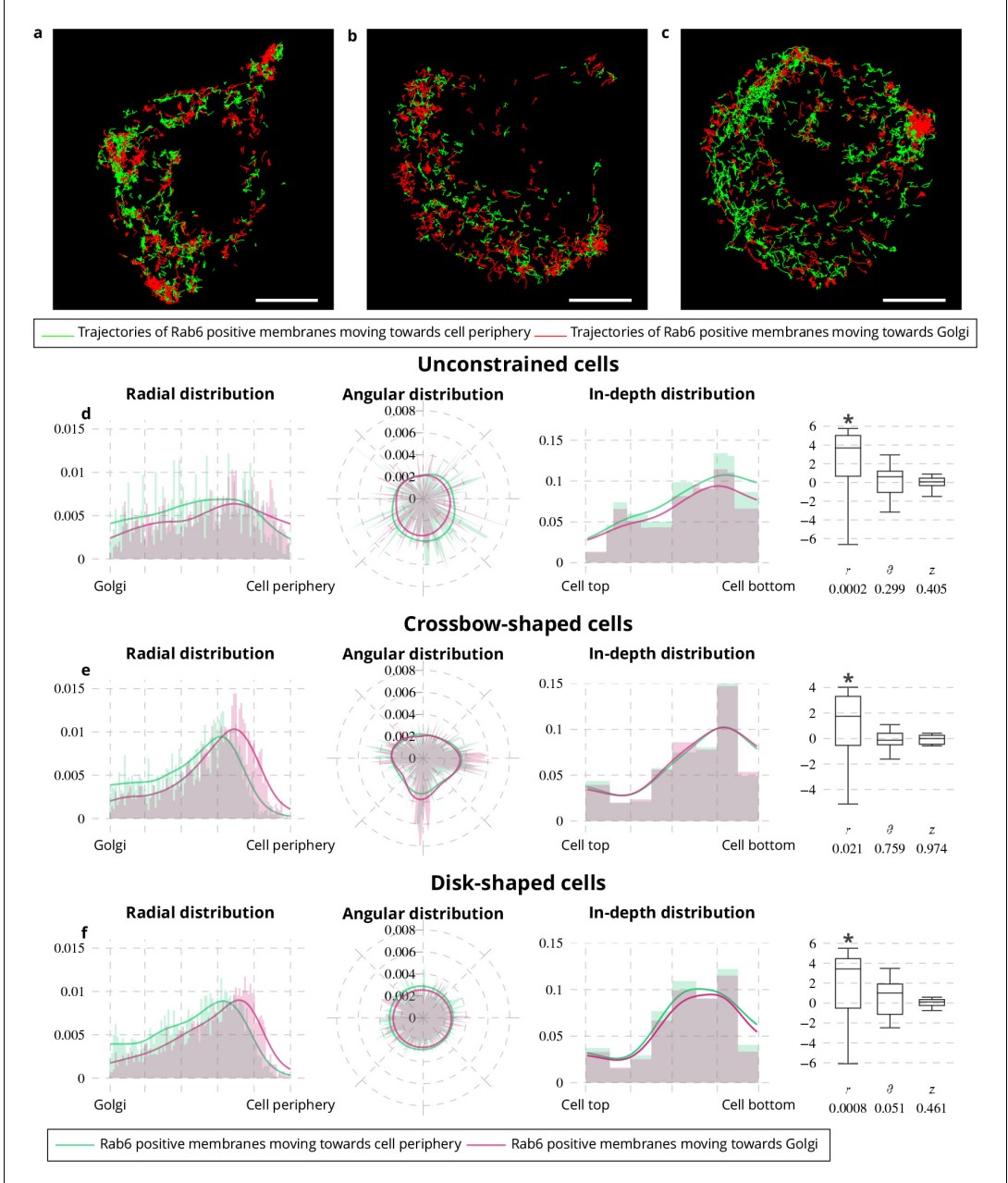

**Figure 3.** Spatial distribution analysis of moving Rab6 proteins. (a–c) Rab6 trajectories moving toward the cell periphery (red trajectories) and toward Golgi (green trajectories) extracted from an image sequence with an unconstrained (a), a crossbow- (b) and a disk- (c) shaped cell. The scale bars correspond to 5 $\mu$m. (d-f) Histograms (bar plots) and densities (lines) of the spatial distribution of Rab6-positive membranes moving toward the cell periphery (green bar plots and lines) or toward the Golgi (pink bar plots and lines) with respect to radius $r$, angle $\theta$ and depth $z$. These distributions come from 18 image sequences with an unconstrained cell (d), 18 image sequences with a crossbow-shaped cell (e) and 22 image sequences with a disk-shaped cell (f). The box and whisker plots of the condition differences of the spatial distribution of moving Rab6-positive membranes with respect to radius $r$, angle $\theta$ and depth $z$ for unconstrained (d), crossbow- (e) and disk- (f) shaped cells are next to the histograms and densities. p values under conditions of one-sided Wilcoxon signed-rank test when considering the condition differences are indicated below the box and whisker plots. A star (*) indicates that the p value is smaller than 0.05.

DOI: https://doi.org/10.7554/eLife.32311.005

statements. To go further in the analysis, we looked at the confinement ratio (*Figure 4a*), the total path length and the lifetime of Rab6 trajectories, conventional dynamical measures used for particle tracking analysis. The combination of these measures with spatial localization is of high interest (*Applegate et al., 2011*; *Tinevez et al., 2017*) and QuantEv provides a good framework to quantitatively analyze and visualize the distribution of these dynamical measures with respect to their intracellular localization. We focus on the radial distribution of Rab6 trajectories from unconstrained and constrained cells as the differences between trajectories moving toward cell periphery and trajectories moving toward Golgi lie in these distributions (see *Figure 3d–f*). Rab6-positive membranes moving toward the cell periphery have a much more direct path than the ones moving toward the Golgi, except near the cell periphery (see *Figure 4b, e*). Consistently, Rab6 positive membranes moving toward the Golgi have longer total path length and lifetime than the ones moving toward the cell periphery, especially when approaching the cell periphery (see *Figure 4c–e*). In summary, this analysis clearly demonstrated that Rab6-positive membranes move predominantly and quite directly from the Golgi to the cell periphery until they enter a docking phase. Then, they mostly go back toward the cell center by following long and indirect trajectories.

## The endosomal recycling compartment organizes Rab11 angular distribution

Rab11-positive recycling membranes originate their journey from the so-called endosomal recycling compartment (ERC). We formulate the assumption that Rab11 positive membranes are uniformly distributed at the membrane plane around the ERC position within the cell, whatever the cell shape is. To test this hypothesis, we used images acquired at the membrane with TIRF microscopy showing Rab11 proteins (see *Appendix 1—figure 1 c–d*). Most labeled membranes of the ERC are not located near the cell surface. However, for each TIRF sequence, one highly inclined wide field image was also acquired, enabling to visually define its location (red disks in *Figure 5a*). To test our assumption, the QuantEv uniformity analysis is applied by considering intensity on segmented regions. The results are shown in *Figure 5a* (blue disks). To have a line of comparison, we also plot

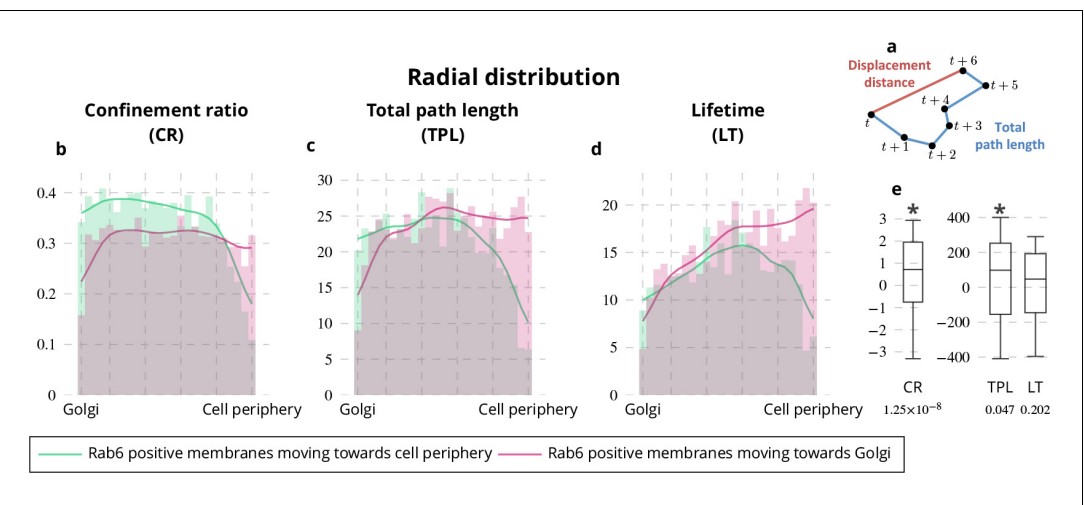

**Figure 4.** Radial distribution of Rab6 dynamic features. (**a**) Illustration of the displacement distance and the total path length of a trajectory. The confinement ratio is defined as the ratio between the displacement distance and the total path length. (**b-d**) Histograms (bar plots) and densities (lines) showing the radial distribution of confinement ratio (**b**), total path length (**c**) and lifetime (**d**) for trajectories of Rab6-positive membranes. These distributions come from the grouping of 18 image sequences with unconstrained cells, 18 image sequences with crossbow-shaped cells and 22 image sequences with disk-shaped cells. (**e**) Box and whisker plots showing the condition differences of the radial distribution of confinement ratio (CR), total path length (TPL) and lifetime (LT) for trajectories of Rab6-positive membranes. p values under conditions of one-sided Wilcoxon signed-rank test when considering the condition differences are indicated below the box and whisker plots. A star (*) indicates that the p value is smaller than 0.05.
DOI: https://doi.org/10.7554/eLife.32311.006

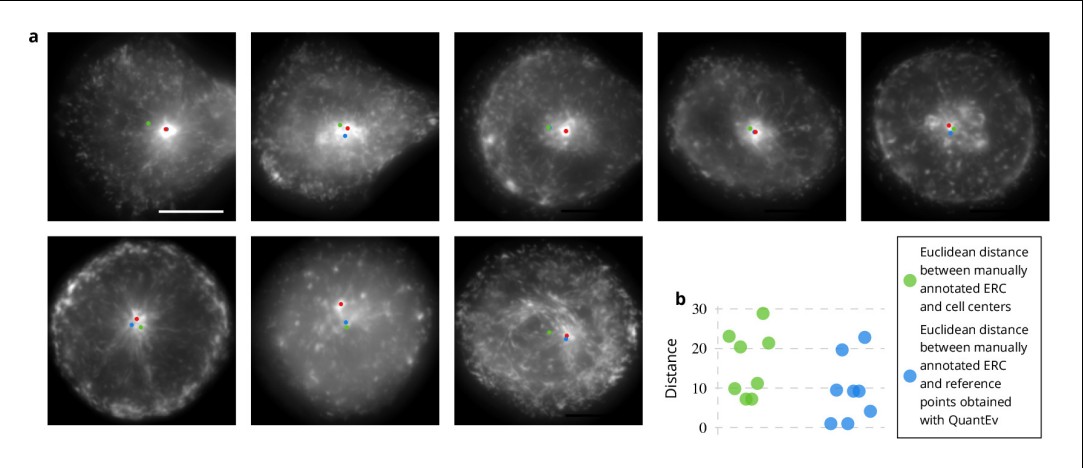

**Figure 5.** Estimation of the endosomal recycling center (ERC) location from the angular distribution of Rab11-positive membranes. (**a**) The red disks correspond to the manual annotation, the blue disks to the point defining the most uniform angular distribution of Rab11-positive membranes and the green disks correspond to the cell centers. These disks are displayed over the average intensity projections of the image sequences showing Rab11-positive membranes. The scale bar corresponds to 5 $\mu$m. (**b**) Euclidean distances between the manually annotated ERC and the cell centers (green disks) or the points giving the most uniform angular distribution (blue disks).
DOI: https://doi.org/10.7554/eLife.32311.007

the cell centers as green disks in *Figure 5a*. Interestingly, the blue disk is close to the red disk for all image sequences except one (second line, middle image in *Figure 5a*). The blue disk is also closer to the red disk than the green disk in seven out of eight image sequences (see *Figure 5a–b*). Although the point that gives the most uniform angular distribution does not strictly coincide with the manually identified ERC, it is sufficiently close to indicate that the Rab11-positive membranes are quite uniformly distributed around the ERC position at the membrane plane whatever the cell shape is. This indicates that the ERC corresponds to the organizing hub of the Rab11 carrier vesicles.

## Joint actin disruption and cell shape influence on Rab11 radial distribution

Applying the QuantEv uniformity analysis at each time step of a sequence allows studying the location stability of the particle emitter or attractor. To test if the estimated ERC location is stationary over time, we computed the Euclidean distance between the reference point estimated at time t = 0 and the points estimated for the next frames. In untreated cells, this distance remains stable (see *Figure 6* green line). We analyzed cells treated with Latrunculin A, which inhibits actin polymerization (see *Appendix 1—figure 1 e–f*). We show that the ERC location is moving away as the drug is affecting the cell (see *Figure 6* blue and orange lines), enlightening the role of cytoskeleton in stabilizing the cellular localization of the ERC. We then acquired image sequences of unconstrained, crossbow-shaped and disk-shaped cells at 10 and 15 min after Latrunculin A addition, and we extracted Rab11 trajectories. The confinement ratio of Rab11 tracks is decreasing with time (see *Figure 7*), which is consistent with actin cytoskeleton being involved in Rab11 vesicle trafficking, as already reported (*Schafer et al., 2014*). The radial distribution of Rab11 vesicles is constantly shifting from the cell periphery to the cell center for unconstrained, crossbow- and disk-shaped cells (see *Figure 8a*). However, before and at drug injection time, we observe significant differences in radial distributions between the three tested conditions (p value = 0.0023, see *Figure 8b*). After Latrunculin A treatment, we progressively observe no difference between the radial distributions, as the actin organization is drastically perturbed. Together, these quantifications allow us to conclude that exocytosis/recycling vesicle trafficking is dependent on both cell shape and actin organization.

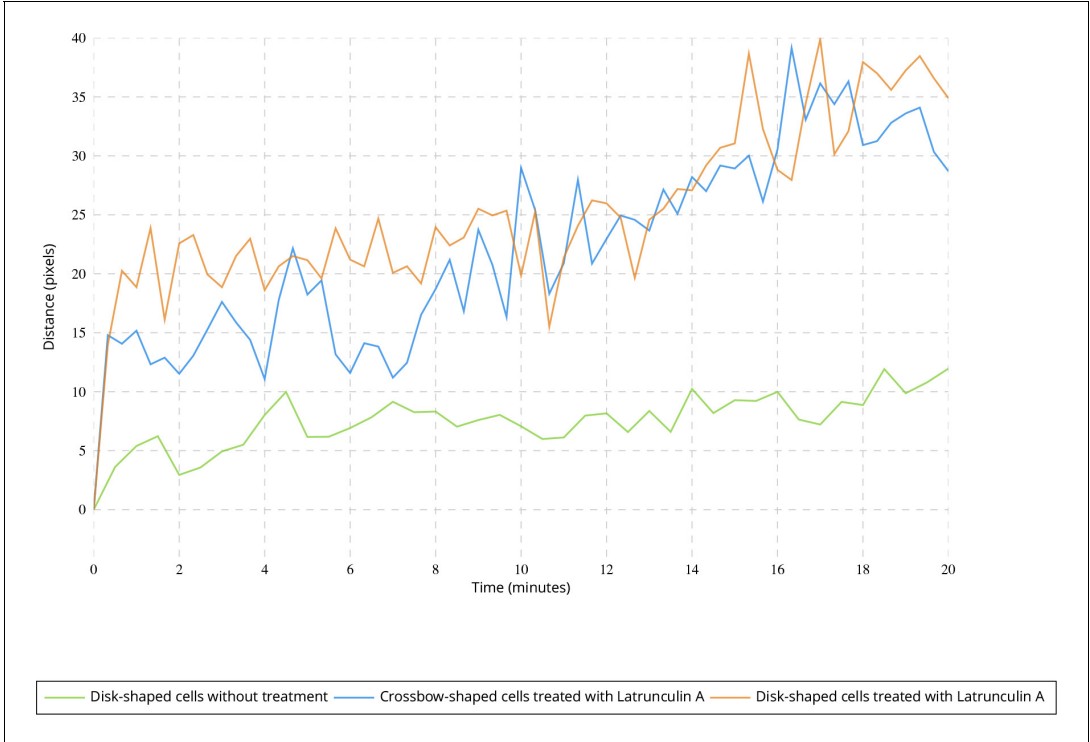

**Figure 6.** Evolution of the point giving the most uniform angular distribution over time. Average Euclidean distance between the point giving the most uniform distribution at time t = 0 and the point estimated at further frames for untreated disk-shaped cells (four image sequences) and cells treated with Latrunculin A (six image sequences for each micro-pattern).

DOI: https://doi.org/10.7554/eLife.32311.008

## Discussion

This article presents a computational framework taking into account cell variability to quantify the distribution of fluorescently labeled proteins. Using dynamical descriptors, detailed insight into dynamical processes is also unraveled and the uniformity analysis allows to localize an organizing region for the observed biological objects.

Additionally to the input image, the user has to define three other inputs that depend on the biological application. First, the user has to decide which coordinate system to use. If the imaged cells are flat as in this study (see *Appendix 1—figure 1a–c*), a cylindrical coordinate system is well suited while a spherical coordinate system will fit better rounded cells. If the user is not familiar with cylindrical or spherical coordinate systems, a classical Cartesian system is also available, even though less suited to intracellular spatial distribution. Finally, QuantEv also allows to analyze the spatial distribution with respect to a reference point or to membrane borders (*Heride et al., 2010*). Once the reference coordinate system is chosen, the user has to define a reference point, typically the particle emitter or attractor, and a reference direction in order to fairly compare cells. For example, in this study, the direction between the Golgi and the cell center were used to define a reference direction for unconstrained and disk-shaped cells while the crossbow principal axis was used for crossbow-shaped cells.

As intensity is proportional to the amount of proteins in fluorescence microscopy, using intensity observed in segmented areas is potentially more informative than binary segmentation masks. However, because of phenomena such as photobleaching, phototoxicity, shading, uneven illumination etc., appropriate normalization procedures within and between images need to be applied. If the user is able to correct for these phenomena, it is preferable to use intensity as weights in QuantEv analysis. Otherwise, intensity weights should be avoided.

Given its genericity, QuantEv can easily be applied to any intracellular event and gives useful insights about their spatial distribution across conditions. From these observations, the user can then

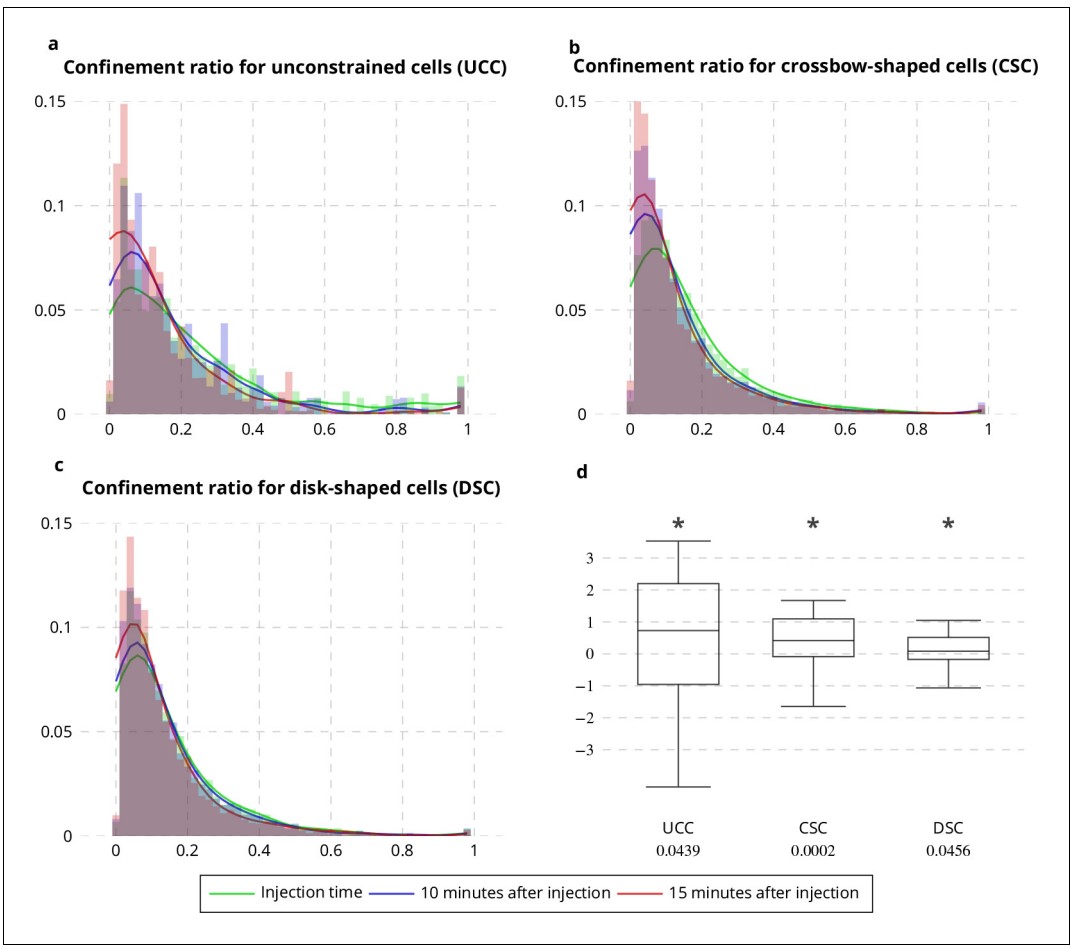

**Figure 7.** Confinement ratio of Rab11-positive membranes with Latrunculin A injection. (**a**-**c**) Histograms (bar plots) and densities (lines) of the confinement ratio of Rab11 positive membranes on unconstrained (**a**), crossbow- (**b**) and disk- (**c**) shaped cells at Latrunculin A Injection Time, 10 and 15 min after injection. (**d**) Box and whisker plots of the corresponding condition differences (five image sequences for unconstrained cells, 10 image sequences for crossbow-shaped cells and nine image sequences for disk-shaped cells). p values under conditions of one-sided Wilcoxon signed-rank test when considering the condition differences are indicated below the box and whisker plots. A star (*) indicates that the p value is smaller than 0.05.
DOI: https://doi.org/10.7554/eLife.32311.009

apply more sophisticated analyses such as mechanistic models of dynamics (*Ponti et al., 2004*; *Jaqaman et al., 2008*) or generative models (*Li et al., 2012*; *Johnson et al., 2015a*, *Johnson et al., 2015b*). QuantEv analysis conclusions can also be the starting point of a new modeling.

We demonstrate with the help of QuantEv that the distributions of Rab6-positive membranes from unconstrained, crossbow- and disk-shaped cells are statistically different. QuantEv also enables to identify the locations where Rab6-positive membranes enter their docking phase. By considering the directions of the moving Rab6-positive membranes, QuantEv allows demonstrating that these membranes first move predominantly and directly toward the cell periphery before reaching their docking phase. They then go back to the cell center in an undirected and long fashion. This intriguing result showing statistically bi-directional movements of Rab6 was reported before. The Rab6-positive vesicles generated at the Golgi membranes are predestined to the cell periphery, in order to deliver their exocytic cargo (*Grigoriev et al., 2007*; *Grigoriev et al., 2011*), which should favor a centrifuge directionality. Our data reconciles this two apparently opposed observations and show for the first time, that a majority of Rab6 vesicles reverses their movement only toward close docking-fusion sites and only during this ultimate phase of docking-fusion.

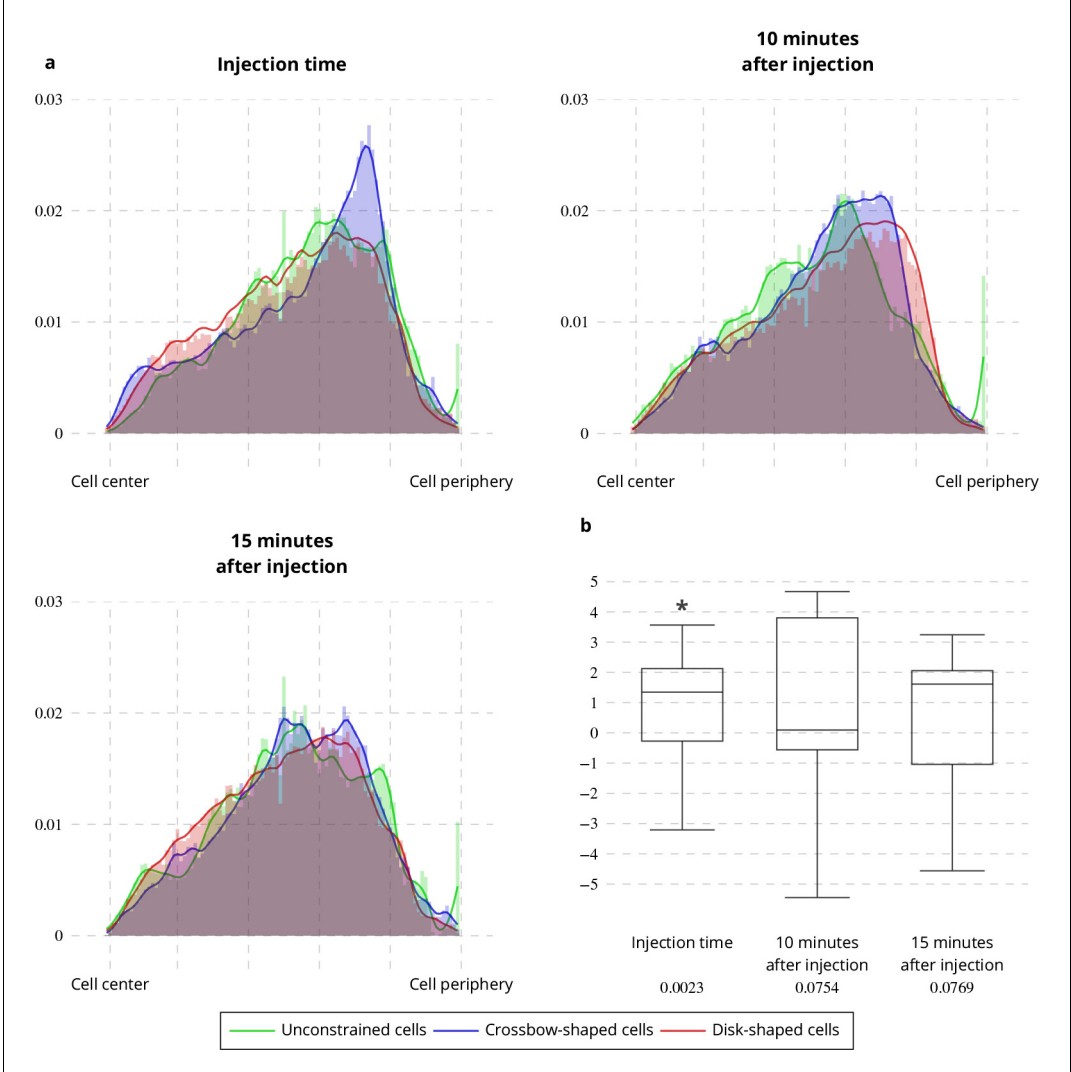

**Figure 8.** Latrunculin A influence on Rab11 radial distribution. (**a**) Histograms (bar plots) and densities (lines) of the radial distribution of Rab11-positive membranes on unconstrained, crossbow- and disk-shaped cells at Latrunculin A Injection Time, 10 and 15 min after injection. (**b**) Box and whisker plots of the condition differences of the radial distribution between unconstrained, crossbow- and disk-shaped cells at Latrunculin A Injection Time, 10 and 15 min after injection. p values under conditions of one-sided Wilcoxon signed-rank test when comparing unconstrained, crossbow- and disk-shaped cells (five image sequences for unconstrained cells, ten image sequences for crossbow-shaped cells and nine image sequences for disk-shaped cells) are indicated below the box and whisker plots.

DOI: https://doi.org/10.7554/eLife.32311.010

QuantEv also demonstrates that Rab11 positive membranes are uniformly distributed around the ERC at the plasma membrane plane. This shows that the ERC represents an organizing hub for the Rab11 carrier vesicles. By applying the QuantEv uniformity analysis along time, we exhibit how the ERC location is affected by actin disruption caused by Latrunculin A injection. The radial distribution analysis of Rab11-positive membranes combined with Latrunculin A injection reveals a dual regulation by cell shape and actin organization on Rab11 trafficking at the plasma membrane, and more generally on the exocytosis/recycling vesicle distribution.

In conclusion, QuantEv has the potential to become a very popular analysis method for dynamics and intracellular event analysis as (i) it is publicly available; (ii) it is fully automated and semi-parametric; (iii) it provides results that are easy to biologically interpret; (iv) it performs a statistical analysis

that takes into account the biological variability over the replicated experiments of a same condition and is efficient with small and large amounts of data. On a biological prospective beyond the two particular models presented here, QuantEv will be of great interest for studies where quantitative and statistical analysis of intracellular membrane or particle behaviors are required, depending on physical and external constraints. For instance, in single-cell experiments performed in microfluidics devices, QuantEv will efficiently provide automation and diversity of statistical analyses in 'one shot', for a relatively small amount of data. Applying QuantEv in multi-cellular systems, in which cell-cell constraints necessarily affect molecular distribution and particle movements will also be of great interest. Finally, in vivo imaging of single-cell intracellular processes in a very confined and constrained environment will benefit from the generic aspect of the QuantEv sensing and measuring of particle spatial distributions, dynamical measures with respect to intracellular localization and cell to cell variability. An Icy plugin and a tutorial are available at http://icy.bioimageanalysis.org/plugin/ QuantEv. A QuantEv analysis module is available on TrackMate and a QuantEv track processor is available in Icy.

## Materials and methods

### Sample preparation

In the first dataset, we use cell lines stably expressing fluorescently tagged proteins in order to minimize the cell-to-cell variability in fluorescence signal. HeLa cells stably expressing fluorescently tagged GFP-RAB6A were previously generated in the Lab at Institut Curie (*Teber et al., 2005*). They were maintained in DMEM supplemented with 10% fetal bovine serum. Cells were then spread onto fibronectin Cytoo chips (Cytoo Cell Architect) 4 to 5 hr before imaging. Cell adhesion on micro-patterns both constrains the cells in terms of lateral movement and averages their size and shape (disk-shaped and crossbow-shaped, Cytoo Cell Architect, $1100\mu$ m$^2$). As a control of patterning effect, the same cell line was grown under the same culture conditions, and spread on regular glass coverslips, 4 to 5 hr before imaging.

For a second set of experiments, wild-type RPE1 cells (hTERT RPE-1 obtained from ATCC collection) were grown in Dulbecco's Modified Eagle Medium, Nutrient Mixture F-12 (DMEM/F12) supplemented with 10% (vol/vol) FCS in six-well plates. RPE1 cells were transiently transfected with plasmids coding for Rab11a-GFP, and Langerin-mCherry using the following protocol: 2 $\mu$g of each DNAs, completed to 100 $\mu$L with DMEM/F12 (FCS free) were incubated for 5 min at room temperature. 6 $\mu$L of X-tremeGENE 9 DNA Transfection Reagent (Roche) completed to 100 $\mu$L with DMEM/ F12 (FCS free) were added to the mix and incubated for further 15 min at room temperature. The transfection mix was then added to RPE1 cells grown 1 day before and incubated further at 37$^o$C overnight. Cells were then spread on regular coverslips or onto fibronectin Cytoo chips (Cytoo Cell Architect) for 4 hr at 37$^o$C with F-12 (with 10% (vol/vol) FCS, 10 mM Hepes, 100 units/ml of penicillin and 100 ug/ml of Strep) before imaging. When specified, 2 mM Latrunculin A (Sigma) was dissolved to 0.02 mM in F-12 DMEM. 300 $\mu$L of culture medium with Latrunculin A (600 nM) was added to establish a final Latrunculin A concentration of 3 $\mu$M.

All cell lines were routinely tested for mycoplasma, using PCR or the MycoAlert Mycoplasma Detection Assay.

### Data acquisition

For Rab6-positive membranes in unconstrained cells, videos were recorded with an epifluorescence video automated system composed of a Ti Eclipse inverted microscope equipped with a 100x objective Plan NA (1.4) and a piezo stage for 3D acquisitions (Nikon, S.A, France). The fluorescence was collected using a 512 $\times$ 512 EM-CCD (Evolve, Photometric, USA) and driven through the Metamorph software (Molecular Devices). 18 series of 120 Z image stacks of 10 frames were recorded at a rate of about 1 stack/s. The volume rendering of one image from this dataset is shown in *Appendix 1—figure 1a*.

For Rab6-positive membranes on micro-patterns, the 488 nm laser of a spinning-disk confocal microscope (Ti Eclipse, Nikon, S.A, France equipped with spinning disk system, a 100x/1.4 oil objective and CoolSnap HQ2 CCD, from Roper Scientific S.A.R.L, France) was used to acquire 3D 380 $\times$ 380 $\times$ 8 stacks at a rate of one stack per second. 18 image sequences with crossbow-shaped cells

and 22 image sequences with disk-shaped cells were acquired. The system was driven by the Metamorph software (Molecular Devices). The volume rendering of two images from this dataset are shown in *Appendix 1—figure 1b–c*.

For the Rab11 dataset, live-cell imaging was performed using simultaneous dual color Total Internal Reflection Fluorescence (TIRF) microscopy. All imagings were performed in full conditioned medium at $37^o$C and 5% CO2 unless otherwise indicated. Simultaneous dual color TIRF microscopy sequences were acquired on a Nikon TE2000 inverted microscope equipped with a 100x TIRF objective (NA = 1.49), an azimuthal TIRF module (Ilas2, Roper Scientifc), an image splitter (DV, Roper Scientific) installed in front of an EMCCD camera (Evolve, Photometrics) that can be bypassed or not, depending on the experimental conditions, as indicated in the text, and a temperature controller (LIS). GFP and m-Cherry were excited with a 488 nm and a 561 nm laser, respectively (100 mW). The system was driven by the Metamorph software (Molecular Devices). Four selected image projections from this data set are shown in *Appendix 1—figure 1d–g*.

### Data availability
We use two datasets in this study that are publicly available on the *iMANAGE* database at https://cid.curie.fr/iManage/standard/login.html with username *public* and password *Welcome!1* in the project entitled QuantEv-Data.

## Event detection and localization
Before applying QuantEv, the intracellular events have to be identified and localized. The Rab6 proteins are extracted from each image sequence by using the C-CRAFT method (*Pécot et al., 2015*) with default parameters, except the p value that ranges from 0.0025 to 0.35 depending on the noise level, available on *Icy* (*de Chaumont et al., 2012*). The Rab11-positive membranes on micro-patterns are segmented at each time point with the ATLAS algorithm (*Basset et al., 2015*) with default parameters, except the p value that ranges from 0.05 to 0.45 depending on the noise level. In both cases, a variance stabilization transform (*Boulanger et al., 2010*) is performed to take into account the Poisson-Gaussian nature of the noise in the CCD sensors. As unconstrained cells are more mobile than cells on micro-patterns, the image sequences showing Rab11-positive membranes in unconstrained cells are not in focus. To correct this phenomenon, a deconvolution method (*Lefkimmiatis et al., 2012*) is first applied to the image sequences. The Rab11 positive membranes are then segmented at each time point with the Bernsen local thresholding method (*Bernsen, 1986*) (radius equal to 15 pixels). Finally, Rab6 and Rab11 trajectories are estimated with the multiple hypothesis tracking method developed by *Chenouard et al. (2013)* with default parameters, available on *Icy* (*de Chaumont et al., 2012*), the combinatorial optimization tracking method developed by *Sbalzarini and Koumoutsakos (2005)* with default parameters, available on *ImageJ* (*Schneider et al., 2012*) and the hybrid approach TrackMate (*Tinevez et al., 2017*) that first connects detected points into short tracks and then links the resulting tracks together, with default parameters, available on Fiji (*Schindelin et al., 2012*). To identify the trajectories estimated with different methods, we use the gated distance (*Chenouard et al., 2014*) defined between two trajectories $\theta_1$ and $\theta_2$ as:

$$d(\theta_1, \theta_2) = \sum_{t=0}^{T} min(||\theta_1(t) - \theta_2(t)||_2, \epsilon), \qquad (1)$$

where $\epsilon$ is the gate. For each image sequence, the gated distance is computed between the trajectories estimated with the three different methods with $\epsilon = 5$ pixels. Only the trajectories for which the gated distance is inferior to 2 pixels for at least two methods are used for the analysis.

## Weighted density estimation for spatial localization
The localization of events needs to be defined on a common coordinate system to compare the experiments. We propose to use the cylindrical coordinate system where only a reference point such as the event emitter or attractor and a reference direction have to be specified by the user. To fairly compare experiments with different cell shapes, we define appropriate distances to obtain normalized densities, that is independent from the cell shape. We illustrate the importance of shape normalization in Appendix 2.

More formally, let us define $\Omega$ the 3D cell support and $\partial\Omega$ the 3D cell surface. Let us consider a set of $N$ sample points associated with intracellular events $S = \{(r_i, \theta_i, z_i, w_i, d_{\theta_i}, d_{z_i}), i \in [1, N]\}$, where $(r_i, \theta_i, z_i)$ denote the spatial cylindrical coordinates. The weight $w_i$ enables to take into account features associated to events such as intensity, track length, confinement ratio... $w_i$ can typically be a function of fluorescence intensity, proportional to the number of molecules observed at a given location. The distance $d_{\theta_i}$ is equal to the Euclidean distance between the coordinate system origin $O \in \Omega$ projected on plane $z_i$ ($O_{z_i}$) and the point $P_{\theta_i, z_i} \in \partial\Omega$ with angle $\theta_i$ at plane $z_i$, such that $d_{\theta_i} = ||P_{\theta_i, z_i} - O_{z_i}||_2^2$. The distance $d_{z_i}$ is equal to the Euclidean distance between the coordinate system origin $O$ and the point $P_{r_i, \theta_i} \in \partial\Omega$ with radius $r_i$ and angle $\theta_i$ such that $d_{z_i} = ||P_{r_i, \theta_i} - O||$. These two distances allow estimating normalized densities that are independent from cell shapes. We propose to estimate three densities defined as follows:

$$f(r) = \frac{1}{Z_{r,\theta}} \sum_{i=1}^{N} G_{\hat{\sigma}_r}(r_i - r) \frac{w_i}{d_{\theta_i}},$$

$$f(\theta) = \frac{1}{Z_{r,\theta}} \sum_{i=1}^{N} H_{\hat{\kappa}}(\theta_i - \theta) \frac{w_i}{d_{\theta_i}}, \tag{2}$$

$$f(z) = \frac{1}{Z_z} \sum_{i=1}^{N} G_{\hat{\sigma}_z}(z_i - z) \frac{w_i}{d_{z_i}},$$

where $G_{\hat{\sigma}}(\cdot)$ is a Gaussian kernel with bandwidth $\hat{\sigma}$, $H_{\hat{\kappa}}$ is a von Mises kernel with concentration $\hat{\kappa}$ such that $H_{\hat{\kappa}}(\theta) = \frac{e^{\hat{\kappa}cos\theta}}{2\pi I_0(\hat{\kappa})}$ and $I_0(\cdot)$ is the Bessel function of order 0. The bandwidths $\hat{\sigma}_r$ and $\hat{\sigma}_z$ are estimated with the Silverman's rule of thumb (*Silverman, 1986*) and $\hat{\kappa}$ is estimated using the robust rule of thumb proposed by *Taylor (2008)*. The normalization constants are defined as follows:

$$Z_{r,\theta} = N \sum_{i=1}^{N} \frac{w_i}{d_{\theta_i}}, \quad Z_z = N \sum_{i=1}^{N} \frac{w_i}{d_{z_i}}. \tag{3}$$

## Density estimation for dynamical features

In case the distribution of dynamical features such as confinement ratio or lifetime with respect to the track localization is to be studied, the weighted densities are defined differently than in the previous section. In this case, histograms are first computed as the averaged dynamic features for each bin. A density estimation is then estimated from the histograms.

Let us consider a set of $T$ tracks associated with spatial coordinates $T = \{(r_i, \theta_i, z_i, m_i), i \in [1, T]\}$, where $(r_i, \theta_i, z_i)$ denote the spatial cylindrical coordinates of the median point of the trajectory $i$ and $m_i$ is a dynamic feature associated to track $i$. The histograms corresponding to the averaged dynamic features for each bin are defined as:

$$h_r(b_r) = \frac{\sum_{i}^{T} \mathbb{1}_{b_r}[r_i] m_i}{\sum_{i}^{T} \mathbb{1}_{b_r}[r_i]},$$

$$h_\theta(b_\theta) = \frac{\sum_{i}^{T} \mathbb{1}_{b_\theta}[\theta_i] m_i}{\sum_{i}^{T} \mathbb{1}\mathbb{1}_{b_\theta}[\theta_i]}, \tag{4}$$

$$h_z(b_z) = \frac{\sum_{i}^{T} \mathbb{1}_{b_z}[z_i] m_i}{\sum_{i}^{T} \mathbb{1}_{b_z}[z_i]},$$

where $b_r \in [1, B_r]$ is a radius bin and $B_r$ is the total number of radius bins, $b_\theta \in [1, B_\theta]$ is a polar bin and $B_\theta$ is the total number of polar bins, $b_z \in [1, B_z]$ is an in-depth bin and $B_z$ is the total number of in-depth bins, and $1_{b_r}[r_i]$ is equal to 1 if $r_i$ is defined in bin $b_r$ and equal to 00 otherwise. Densities are then estimated from the histograms as follows:

$$f_d(r) = \sum_{i=1}^{B_r} G_{\hat{\sigma}_r}(h_r(b_i) - r),$$

$$f_d(\theta) = \sum_{i=1}^{B_\theta} H_{\hat{\kappa}}(h_\theta(b_i) - \theta), \qquad (5)$$

$$f_d(z) = \sum_{i=1}^{B_z} G_{\hat{\sigma}_z}(h_z(b_i) - z),$$

where $G_{\hat{\sigma}}(\cdot)$ is a Gaussian kernel with bandwidth $\hat{\sigma}$, $H_{\hat{\kappa}}$ is a von Mises kernel with concentration $\hat{\kappa}$. The bandwidths $\hat{\sigma}_r$ and $\hat{\sigma}_z$ are estimated with the Silverman's rule of thumb (*Silverman, 1986*) and $\hat{\kappa}$ is estimated using the robust rule of thumb proposed by *Taylor (2008)*.

## Statistical procedure

Quantitative comparison between different conditions is mandatory to analyze biological data. In most computational biology studies, data from different experiments corresponding to the same condition are pooled together (*Schauer et al., 2010*; *Merouane et al., 2015*). This usual procedure enables to add statistical power when comparing two conditions. Therefore, it is especially useful when few data are available. Unfortunately, pooling data together presents two main drawbacks. First, if large amounts of data are available, the opposite problem arises and the statistical tests may become significant for every comparison (*Olivier and Walter, 2015*). One solution is to down sample the data, but the amount of down sampling becomes another issue. Second, pooling data together for one condition partially hides the variability between the replicated experiments for this condition. As an example, let us consider a study aimed at analyzing the effects of a drug on a sample of normal individuals. To evaluate the drug efficiency, a comparison between normal individuals and individuals that were administered the drug is conducted. Let us assume that the drug is effective on half the individuals. Consequently, normal individuals are compared to a mix of normal individuals and individuals with the drug effects. This comparison should not be statistically significant as the drug is not efficient on all individuals. However, the effects on the individuals for which the drug is efficient might hide the fact that it is not efficient on all individuals if all the data are pooled together. In what follows, we propose to compute a distance between all experiments instead of a distance between conditions. The idea is demonstrated in Appendix 3 and validated on synthetic image sequences (see *Appendix 1—figure 1c–d* and Appendix 3).

## Distance between densities

We propose to compute the earth mover's distance (also known as the Kantorovich-Rubinstein or the first order Wasserstein distance) between every replicate of every condition to apply a statistical test. This transport-based distance demonstrated its efficiency for other studies on cell phenotypes (*Wang et al., 2013*; *Basu et al., 2014*). The discrete Earth Mover's Distance (EMD) between two uni-dimensional distributions is simply defined as the sum of the absolute differences between their cumulated distribution functions (*Rubner et al., 2000*):

$$EMD(f^1, f^2) = \sum_{i=1}^{K} |F^1(i) - F^2(i)|, \qquad (6)$$

where $F^1$ and $F^2$ are the cumulated distribution functions of $f^1$ and $f^2$. Although the EMD depends on the number of bins $K$, EMD proportions are kept intact when the number of bins is high enough as shown in Appendix 4. For the angular distribution, the Circular Earth Mover's Distance (CEMD) (*Rabin et al., 2011*) is defined as:

$$\mathrm{CEMD}\left(f^1, f^2\right) = \min_{k \in \{1, \cdots, K\}} \sum_{i=1}^{K} |Q_k^1(i) - Q_k^2(i)|, \tag{7}$$

with

$$Q_k(i) = \begin{cases} \sum_{j=k}^{i} f(j) & \text{if } i \geq k, \\ \sum_{j=k}^{K} f(j) + \sum_{j=1}^{i} f(j) & \text{if } i < k. \end{cases} \tag{8}$$

## Difference between conditions

The EMD and CEMD enable to compute a distance between two single experiments for the radial, angular and in-depth densities. The distances between the replicates of one condition and the replicates of the other condition(s) give an idea about the difference between the conditions. However, a baseline distance is also needed to state if the difference is random or significant. Therefore, two distances are defined for each experiment and each density:

- the *intra-condition distance*: average distance between the density and all the other densities for the same condition;
- the *inter-condition distance*: average distance between the density and all the other densities from the other condition(s).

Considering more than two groups does not change the intra-condition distance and only expands the inter-condition distance to more than one group. We define as the *condition difference* the difference between the inter-condition distance and the intra-condition distance. If the *condition difference* is high, the conditions are different.

## Statistical test

A statistical test is applied on the difference distance to state if the observed conditions are significantly different. A non-parametric statistical test is better suited as there is no underlying model for the condition difference. In addition, a negative condition difference implies that the current experiment is closer to the replicated experiments of the other condition than the replicated experiments of the same condition. Consequently, the condition difference has to be positive if the conditions are different. For those two reasons, we propose to use the one-sided non-parametric Wilcoxon signed-rank test on the condition differences for all experiments to state if two conditions are statistically different.

## Analysis of uniform distribution of events

In case we focus on the intracellular events assumed to be uniformly distributed around a given biological object, for example the events emitter, QuantEv allows us to estimate a location for this traffic-organizing component. This source location is then defined as the reference point with the most uniform angular distribution. It is established that the maximum entropy corresponds to the most uniform distribution. Consequently, the reference point $O^*$ is defined as the location that maximizes the entropy:

$$O^* = \max_{O \in \Omega} - \sum_{i=1}^{N} f(\theta_i) \log f(\theta_i). \tag{9}$$

The most straightforward way to find this point is to estimate the entropy map that gives, for each point in $\Omega$, the entropy value computed with the current point used as the reference center. We also propose to use the bisection method to speed up the computation (about sixty times faster than the entropy map computation, see *Appendix 5—table 1*). A uniformity analysis conducted on simulations is presented in Appendix 6. The entropy criterion can be extended to detect multiple organizing components if needed.

## Code availability

The jar file of the QuantEv Icy plugin is available at http://icy.bioimageanalysis.org/plugin/QuantEv. The jar file of the QuantEv track processor is available at http://icy.bioimageanalysis.org/plugin/ QuantEv _(track_processor). The source codes can be extracted from the jar files. The QuantEv analysis module for TrackMate is available on GitHub (*Pécot, 2018*; copy archived at https://github. com/elifesciences-publications/QuantEvForTrackMate). These codes are released under the GNU Affero General Public License v3.0.

## Acknowledgements

This work was supported by the France-BioImaging infrastructure (ANR-10-INBS-04). The Cell and Tissue Imaging Facility at Institut Curie is a member of France-BioImaging. We thank Sabine Bardin for regularly testing mycoplasma infection and for the sample preparation of experiments with Rab6 proteins. We thank Jean-Yves Tinevez for his help to implement the QuantEv analysis module for TrackMate and the Icy track processor.

## Additional information

### Funding

| Funder | Grant reference number | Author |
|---|---|---|
| France-BioImaging | ANR-10-INBS-04 | Charles Kervrann |

The funders had no role in study design, data collection and interpretation, or the decision to submit the work for publication.

### Author contributions

Thierry Pécot, Conceptualization, Data curation, Software, Formal analysis, Validation, Investigation, Visualization, Methodology, Writing—original draft; Liu Zengzhen, Resources; Jérôme Boulanger, Formal analysis, Investigation, Methodology, Writing—review and editing; Jean Salamero, Resources, Funding acquisition, Investigation, Writing—review and editing; Charles Kervrann, Formal analysis, Supervision, Investigation, Methodology, Writing—review and editing

### Author ORCIDs

Thierry Pécot (iD) http://orcid.org/0000-0003-0772-9753
Charles Kervrann (iD) http://orcid.org/0000-0001-6263-0452

### Decision letter and Author response
Decision letter https://doi.org/10.7554/eLife.32311.040
Author response https://doi.org/10.7554/eLife.32311.041

## Additional files

### Supplementary files
• Transparent reporting form
DOI: https://doi.org/10.7554/eLife.32311.011

### Data availability
Image sequences have been deposited in the Institut Curie database, available at https://cid.curie.fr/ iManage/standard/login.html

The following dataset was generated:

| Author(s) | Year | Dataset title | Dataset URL | Database, license, and accessibility information |
|---|---|---|---|---|
| Pecot T | 2017 | QuantEv-Data | https://cid.curie.fr/iMa- | login: public - |

nage/standard/login.
html

password: Welcome!
1

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

## Appendix 1

DOI: https://doi.org/10.7554/eLife.32311.012

### Datasets

An example for each condition of the two datasets used in this study is shown in *Appendix 1—figure 1*.

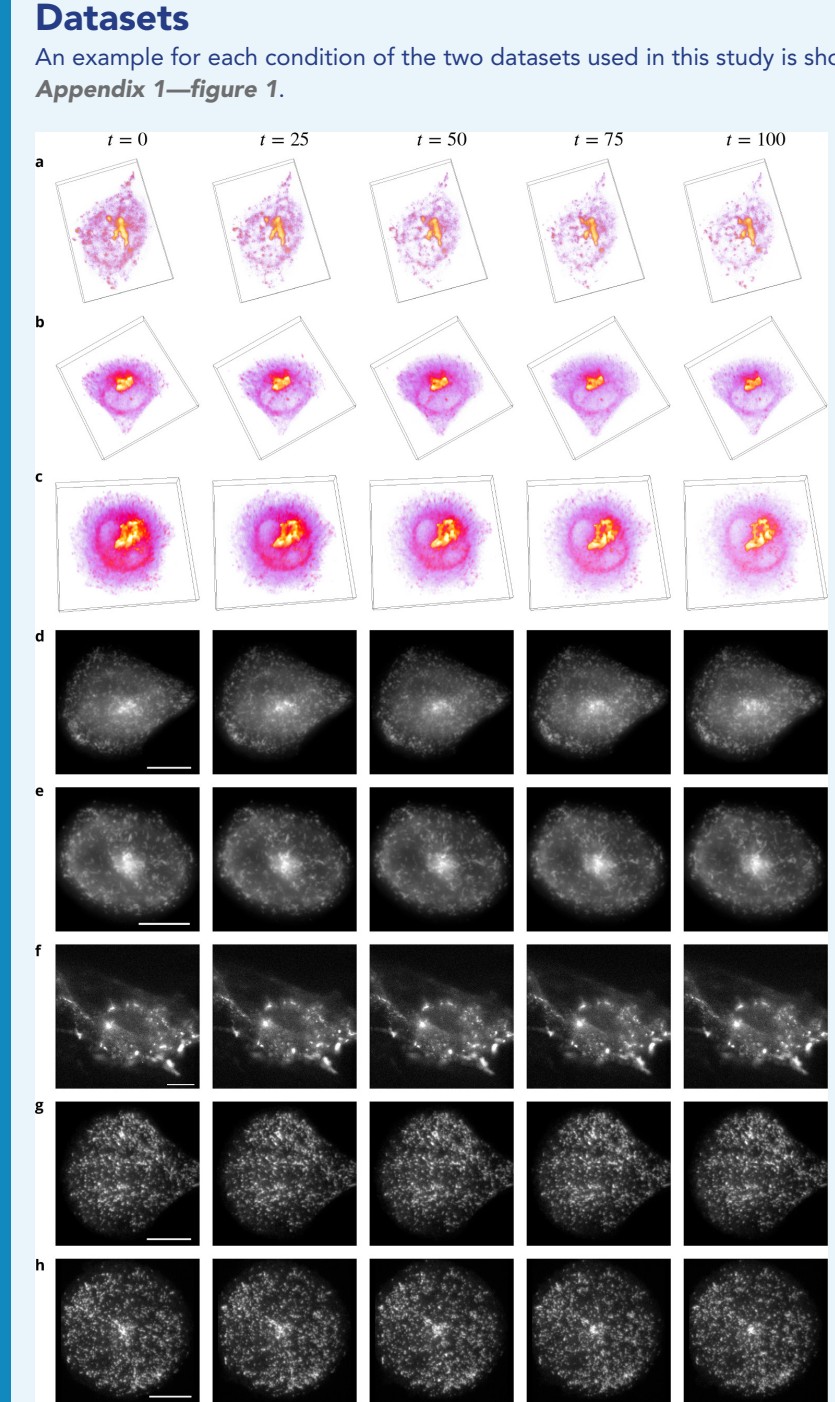

**Appendix 1—figure 1.** Example images from the datasets. (**a-c**) Volume renderings of fluorescent images taken from three sequences showing Rab6 proteins in an unconstrained (**a**), a crossbow- (**b**) and a disk- (**c**) shaped cell. (**d–e**) Fluorescent images taken from two sequences showing Rab11 proteins in a crossbow- (**d**) and a disk- (**e**) shaped cell. (**f–h**) Fluorescent images taken from three sequences showing Rab11 proteins treated with Latrunculin A in an unconstrained (**f**), a crossbow- (**g**) and a disk- (**h**) shaped cell. In Figs **d–h**,

the intensity over the planes is averaged, a gamma correction is applied for a better visualization and the scale bars correspond to 5 $\mu$m.
DOI: https://doi.org/10.7554/eLife.32311.013

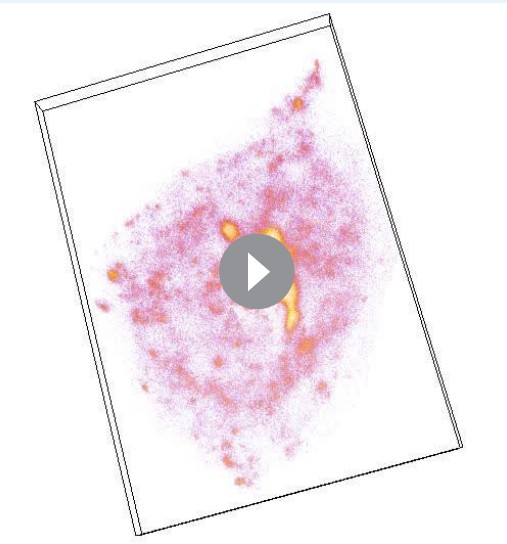

**Appendix 1—figure 1—video 1.** Rab6 proteins in an unconstrained cell.     Volume rendering of fluorescent images showing Rab6 proteins in an unconstrained cell corresponding to *Appendix 1—figure 1a*.
DOI: https://doi.org/10.7554/eLife.32311.014

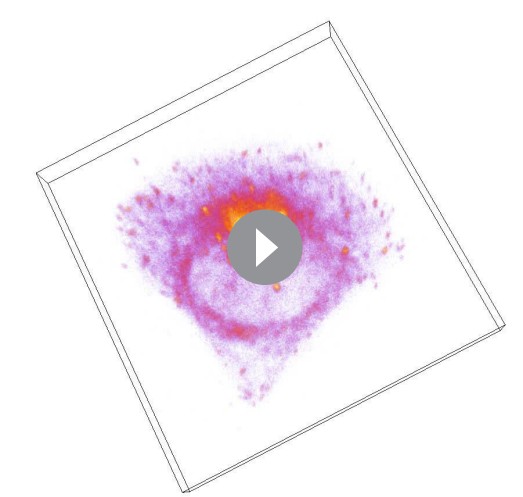

**Appendix 1—figure 1—video 2.** Rab6 proteins in a crossbow-shaped cell.     Volume rendering of fluorescent images showing Rab6 proteins in a crossbow-shaped cell corresponding to *Appendix 1—figure 1b*.
DOI: https://doi.org/10.7554/eLife.32311.015

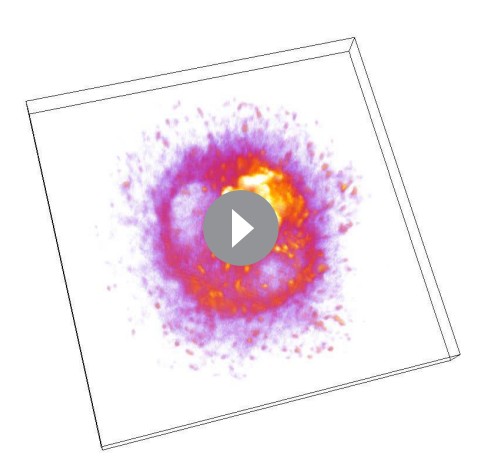

**Appendix 1—figure 1—video 3.** Rab6 proteins in a disk-shaped cell.     Volume rendering of fluorescent images showing Rab6 proteins in a disk-shaped cell corresponding to *Appendix 1—figure 1c*.
DOI: https://doi.org/10.7554/eLife.32311.016

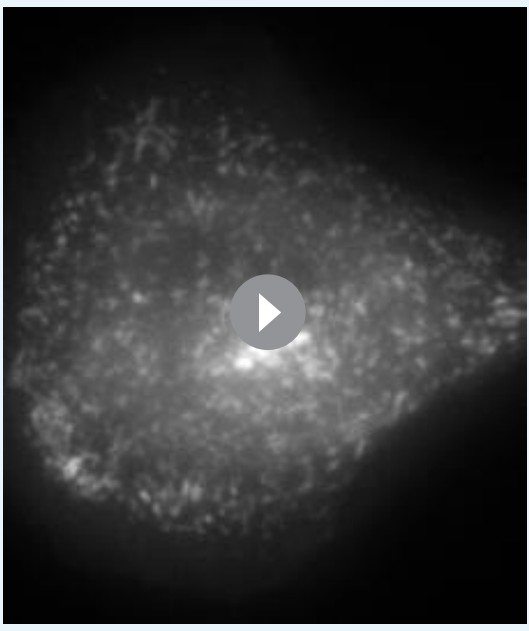

**Appendix 1—figure 1—video 4.** Rab11 proteins in a crossbow-shaped cell.     Image sequence showing Rab11 proteins in a crossbow-shaped cell corresponding to *Appendix 1—figure 1d*.
DOI: https://doi.org/10.7554/eLife.32311.017

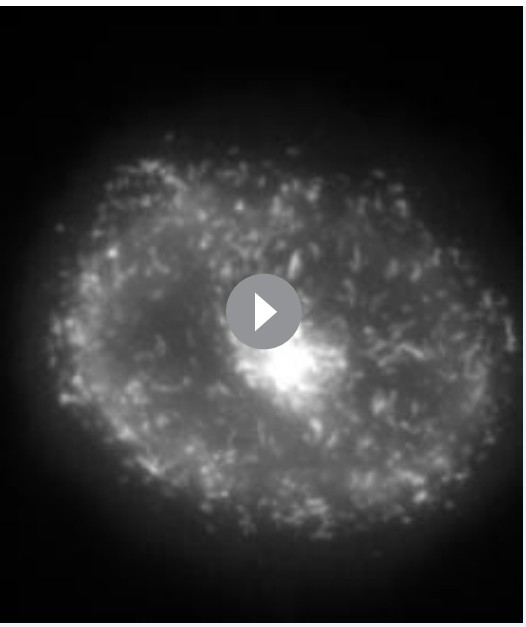

**Appendix 1—figure 1—video 5.** Rab11 proteins in a disk-shaped cell. Image sequence showing Rab11 proteins in a disk-shaped cell corresponding to *Appendix 1—figure 1e*.
DOI: https://doi.org/10.7554/eLife.32311.018

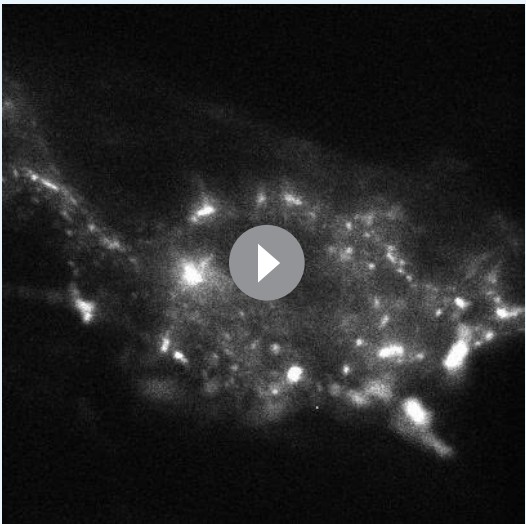

**Appendix 1—figure 1—video 6.** Rab11 proteins treated with Latrunculin A in an unconstrained cell. Image sequence showing Rab11 proteins treated with Latrunculin A in an unconstrained cell corresponding to *Appendix 1—figure 1f*.
DOI: https://doi.org/10.7554/eLife.32311.019

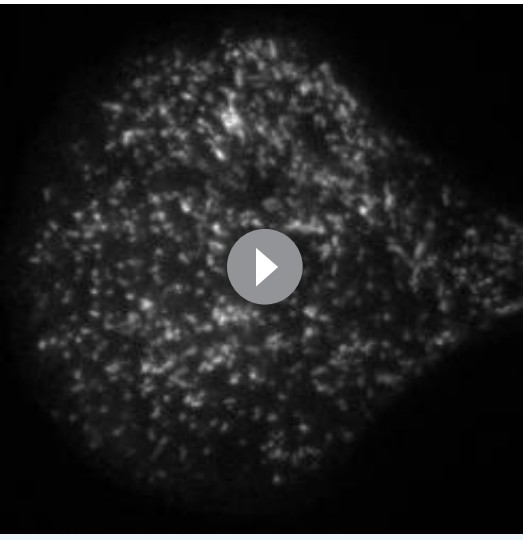

**Appendix 1—figure 1—video 7.** Rab11 proteins treated with Latrunculin A in a crossbow-shaped cell.　　Image sequence showing Rab11 proteins treated with Latrunculin A in a crossbow-shaped cell corresponding to *Appendix 1—figure 1g*.
DOI: https://doi.org/10.7554/eLife.32311.020

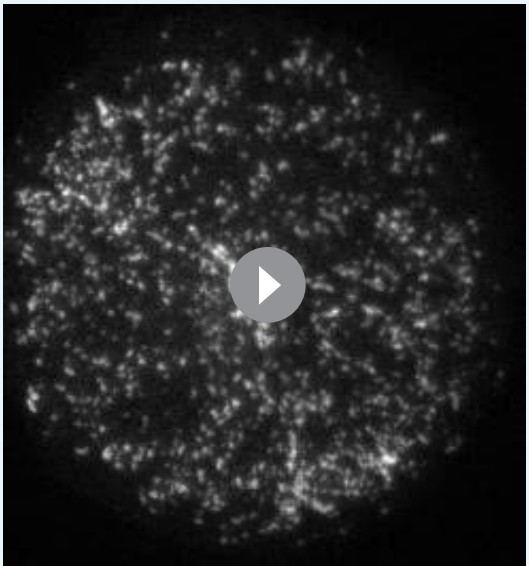

**Appendix 1—figure 1—video 8.** Rab11 proteins treated with Latrunculin A in a disk-shaped cell.　　Image sequence showing Rab11 proteins treated with Latrunculin A in a disk-shaped cell corresponding to *Appendix 1—figure 1h*.
DOI: https://doi.org/10.7554/eLife.32311.021

The four different scenarios of simulated images that were generated to evaluate QuantEv performance are shown in *Appendix 1—figure 2*.

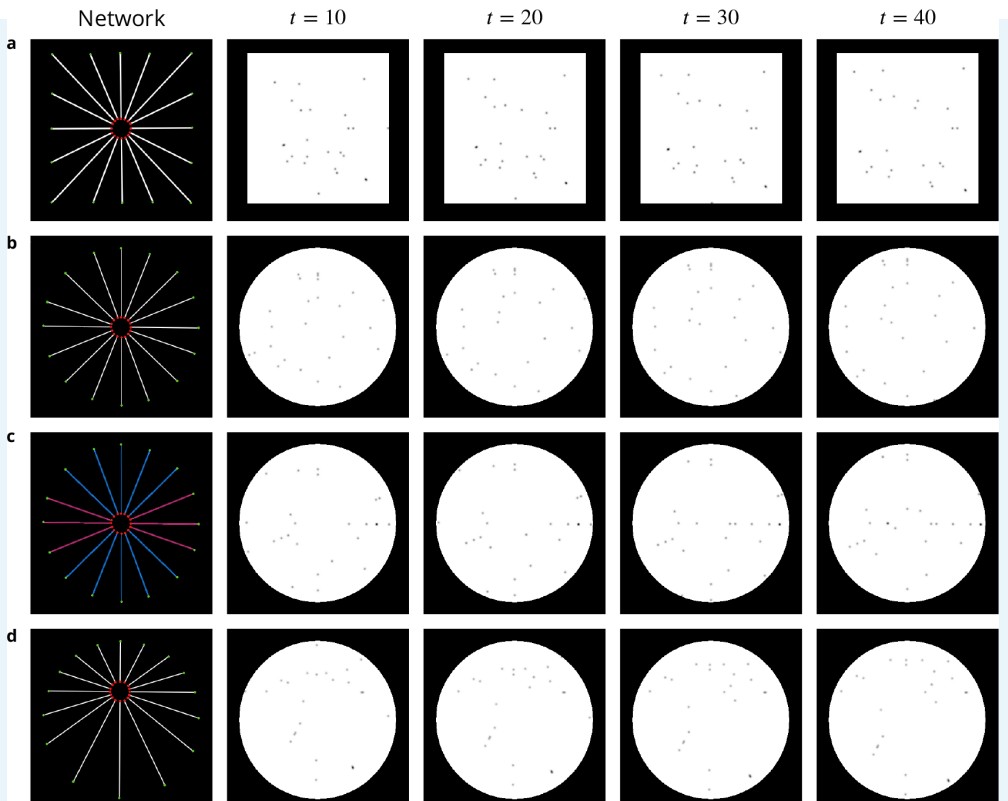

**Appendix 1—figure 2.** Image sequences simulated to evaluate QuantEv performance. First column: networks used to generate four image sequences. The particle origins are labeled as red disks while particle destinations appear as green disks. Particles for the sequences **a**, **c** and **d** are uniformly distributed over the different paths. Particles for the sequence **c** are distributed with a probability equal to 0.1 over the pink paths and with a probability equal to 0.04 over the blue paths. Images corresponding to time t=10, t=20, t=30 and t=40 taken from one simulated image sequence for each network are illustrated in columns 2 to 5.

DOI: https://doi.org/10.7554/eLife.32311.022

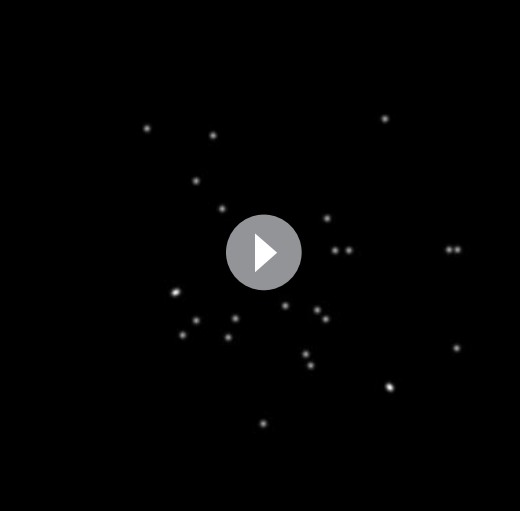

**Appendix 1—figure 2—video 1.** Simulation of uniformly distributed particles in a square-shaped cell.     Image sequence simulated with particles uniformly distributed over 16 paths on a square-shaped cell, corresponding to *Appendix 1—figure 2a*.

DOI: https://doi.org/10.7554/eLife.32311.023

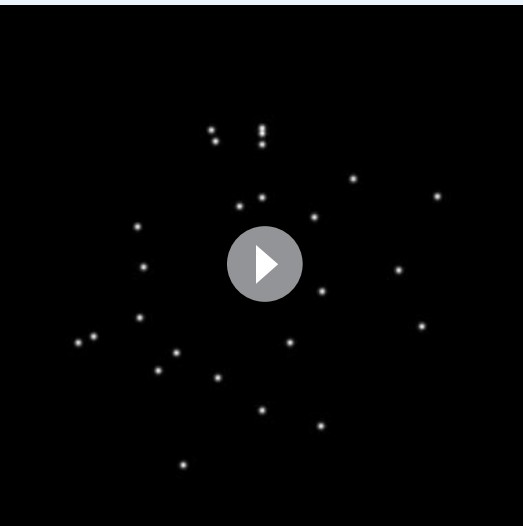

**Appendix 1—figure 2—video 2.** Simulation of uniformly distributed particles in a disk-shaped cell.     Image sequence simulated with particles uniformly distributed over 16 paths on a disk-shaped cell, corresponding to *Appendix 1—figure 2b*.

DOI: https://doi.org/10.7554/eLife.32311.024

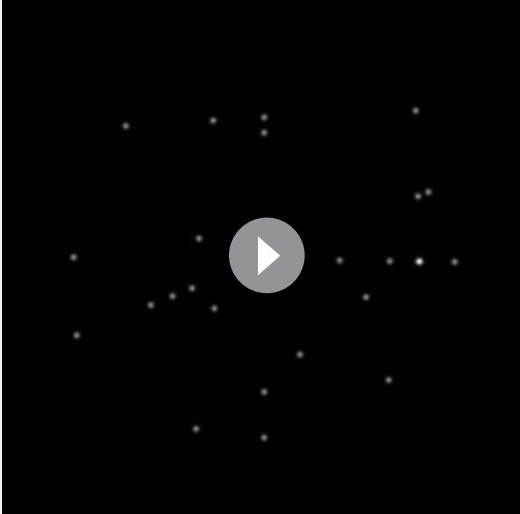

**Appendix 1—figure 2—video 3.** Simulation of isotropically distributed particles in a disk-shaped cell.     Image sequence simulated with particles distributed over 16 paths with two different probabilites on a disk-shaped cell, corresponding to *Appendix 1—figure 2c*.

DOI: https://doi.org/10.7554/eLife.32311.025

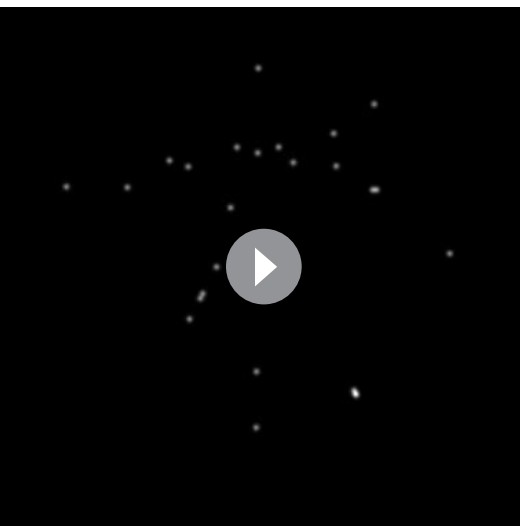

**Appendix 1—figure 2—video 4.** Simulation of uniformly distributed particles around an uncentered emitter in a disk-shaped cell.     Image sequence simulated with particles uniformly distributed over 16 paths on a disk-shaped cell, around an emitter located in the upper part of the cell, corresponding to *Appendix 1—figure 2d*.
DOI: https://doi.org/10.7554/eLife.32311.026

## Appendix 2

DOI: https://doi.org/10.7554/eLife.32311.027

### Sensitivity to cell shape

The cell shape influences the spatial distribution of intracellular events. The distances $d_\theta$ and $d_z$ were introduced to compute a distribution that is invariant from the cell shape (see Section *Weighted density estimation*). To quantify the cell shape influence and to evaluate the pertinence of the normalization with distances, we generate image sequences with vesicles trafficking on a square-shaped region. In these simulations, vesicles are uniformly distributed over 16 different paths and are moving from the cell center to the cell periphery (see *Appendix 1—figure 2a*). As the cell is square-shaped, the vesicles moving to the cell corners travel a longer distance than the other vesicles so the number of vesicles on these paths is higher. Consequently, the spatial distribution of vesicles is not uniform over the radius and angle ranges (see purple histograms in *Appendix 2—figure 1*). Nevertheless, the vesicles are generated over the paths with an equal probability in the simulations, meaning that the distribution over the different paths is uniform. By weighting the distribution of spatial coordinates with the distance between the cell center and the cell periphery, the shape dependence is accurately corrected as shown in the green histograms of *Appendix 2—figure 1*. Although other registration approaches are more powerful (*Zhao and Murphy, 2007*; *Peng and Murphy, 2011*), the results shown in *Appendix 2—figure 1* demonstrate that these sophisticated methods are not needed in our case.

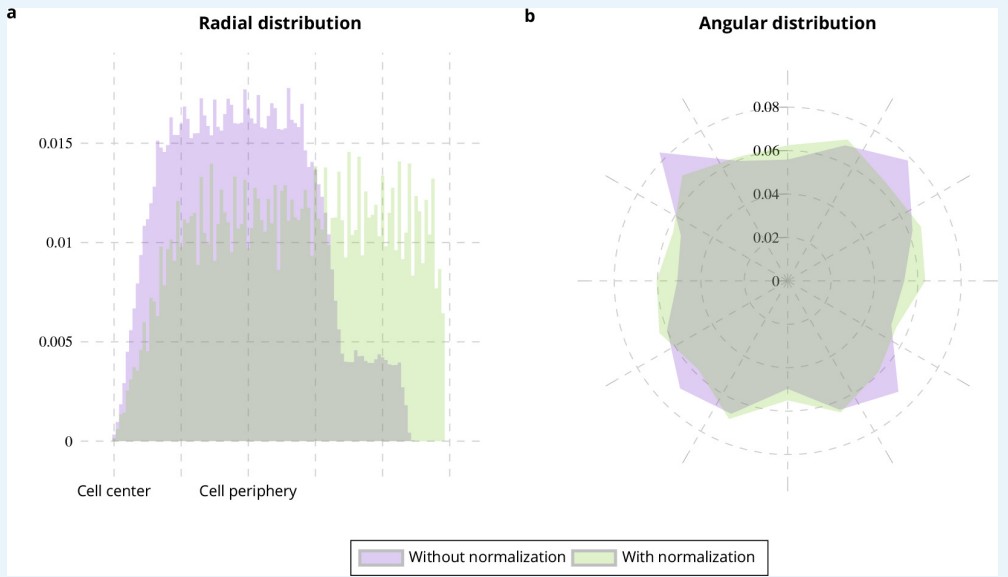

**Appendix 2—figure 1.** Cell shape normalization. Radial (**a**) and angular (**b**) distributions of particles moving on square-shaped cells from 10 simulated image sequences (*Appendix 1—figure 2a*) with (green histograms) and without normalization (purple histograms) with respect to the distance between the cell center and the cell periphery.
DOI: https://doi.org/10.7554/eLife.32311.028

# Appendix 3

DOI: https://doi.org/10.7554/eLife.32311.029

## Statistical analysis

To evaluate the effect of pooling data together on the statistical analysis, 20 image sequences with uniform distribution over the paths (*Appendix 1—figure 2b*) and 10 image sequences with isotropic distribution over the paths (*Appendix 1—figure 2c*) are generated. Four groups are then defined from these simulations:

- group #1: 10 image sequences with uniform distribution;
- group #2: 10 other image sequences with uniform distribution;
- group #3: 10 image sequences with isotropic distribution (six paths with a probability equal to $0.1$ and 10 paths with a probability equal to $0.04$);
- group #4: five image sequences with uniform distribution and five image sequences with isotropic distribution.

The analysis of variance (ANOVA), usual method for biological studies, is compared to the QuantEv statistical approach. For the ANOVA analysis, the vesicle mass centers are extracted from the simulations and the pair $(r, \theta)$ is used to compare two groups. The intensity observed in the segmented vesicles is used for the QuantEv approach. For both methods, several amounts of data are considered: from 1% to 100% data for the ANOVA analysis; from 2 vs. 2 to 10 vs. 10 image sequences for QuantEv.

With the ANOVA analysis on pooled data, the *p*-values are low with a small amount of data when comparing groups #1 and #3 (see *Appendix 3—figure 1a*). But they also start to be low when comparing groups #1 and #2 for an amount of data that reaches about 50% (see *Appendix 3—figure 1a*). These results indicate that there is a gradient of *p*-values consistent with actual differences between the spatial distributions. However, the values lead to a significant difference between all groups (see *Appendix 3—figure 1a*). It demonstrates that it is difficult to deal with pooled data when the amount of data is high. When comparing groups #1 and #4, there should not be any statistical difference, as group #4 is constituted of particles with different distributions. But the ANOVA analysis on the pooled data is not able to grasp this variability between replicated experiments of a same condition and the *p*-values are low with a small amount of data (about 5%, see *Appendix 3—figure 1a*).

The QuantEv statistical approach does not lead to any statistical difference for radius for the three comparisons (see *Appendix 3—figure 1b*), a result that is consistent with the data. By using QuantEv, it turns out that angular distributions are statistically different when comparing groups #1 and #3 while they are not for the two other comparisons (see *Appendix 3—figure 1b*). These experiments demonstrate that the QuantEv statistical approach is not disturbed by large amounts of data because it considers the distributions over the sequences. They also demonstrate that QuantEv takes into account the variability between replicated experiments of a same condition as the comparison involving groups #1 and #4 does not conclude to any statistical difference.

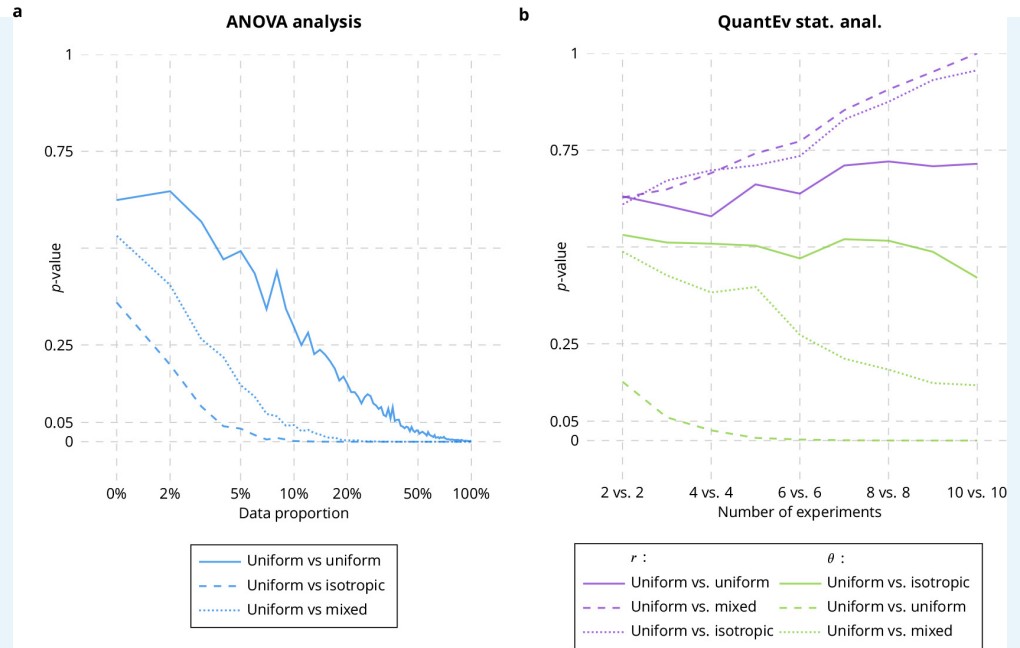

**Appendix 3—figure 1.** Comparison between ANOVA and QuantEv. *p*-values obtained with ANOVA statistical analysis (**a**) and QuantEv statistical analysis (**b**) when considering the spatial distribution of particles uniformly distributed (*Appendix 1—figure 2b*), isotropically distributed (*Appendix 1—figure 2c*) and a mix of uniformly and isotropically distributed particles over 16 paths.

DOI: https://doi.org/10.7554/eLife.32311.030

## Appendix 4

DOI: https://doi.org/10.7554/eLife.32311.031

### Binning influence on earth mover's distance

The QuantEv statistical procedure relies on the circular and regular earth mover's distances, computed from densities defined from binned data. To evaluate the effect of binning on these distances, we computed the Earth Mover's Distances (EMD) between $cos(x)$ and $cos(2x)$ (purple line in *Appendix 4—figure 1a*) and between $sin(x)$ and $cos(2x)$ (green line in *Appendix 4—figure 1a*) with a number of bins ranging from 1 to 500. As expected from (*Bernsen, 1986*), the EMD increases with the number of bins. However, the proportion between distances remains quite constant when the number of bins is higher than 25 (see *Appendix 4—figure 1b*). Consequently, if the EMD between two distributions $f$ and $g$ is higher than the EMD between two distribution $f$ and $h$ for a given number of bins $n_1$, the EMD between $f$ and $g$ is also higher than the EMD between $f$ and $h$ for a number of bins $n_2$. As the statistical test used in the QuantEv framework (one-sided Wilcoxon signed-rank test) is based on the ranks of the difference between inter-EMD and intra-EMD, the result of the test is not affected by the number of bins.

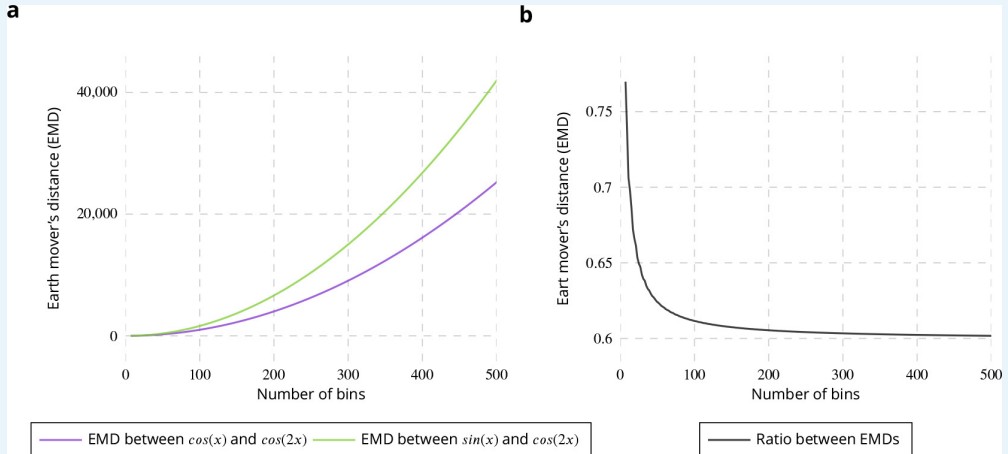

**Appendix 4—figure 1.** Evaluation of binning influence on earth mover's distance. (**a**) Earth Mover's Distance (EMD) between $cos(x)$ and $cos(2x)$ (purple curve) and between $sin(x)$ and $cos(2x)$ (green curve). (**b**) Ratio between the two EMDs shown on left plot.
DOI: https://doi.org/10.7554/eLife.32311.032

# Appendix 5

DOI: https://doi.org/10.7554/eLife.32311.033

## Processing time

QuantEv computation time to process images with different size and object coverage is shown in *Appendix 5—table 1*.

**Appendix 5—table 1.** QuantEv processing time.

| Image size | Object | Histogram and | Point with most uniform distribution | |
|---|---|---|---|---|
| | Coverage | Density computation | Bisection method | Entropy map |
| 256 × 256 | 10% | 0.5 s | 42.1 s | 8min45s |
| 256 × 256 | 60% | 0.6 s | 2min31s | 47min16s |
| 512 × 512 | 10% | 1.6 s | 2 min 20 s | 2h11min |
| 512 × 512 | 60% | 1.7 s | 11 min 48 s | 11 hr 3 min |

DOI: https://doi.org/10.7554/eLife.32311.034

# Appendix 6

DOI: https://doi.org/10.7554/eLife.32311.035

## Uniform distribution of events

To evaluate the QuantEv uniformity analysis, we simulate 10 image sequences with particles uniformly distributed over the different paths on a network for which the origin is not centered in the image (*Appendix 1—figure 2d*). *Appendix 6—figure 1a* shows the entropy map obtained for one simulation. *Appendix 6—figure 1b* shows the different reference points estimated over the ten simulations as green disks. These results are not perfect, as the reference centers are not estimated to be located at the exact particle emitter location. However, if the particles are distributed with equal probability on all paths, this does not imply that the actual number of generated particles is the same on all paths so the estimation cannot be perfect. The estimated reference points for these simulations are close to the particle emitter location, which demonstrates the potential of this approach.

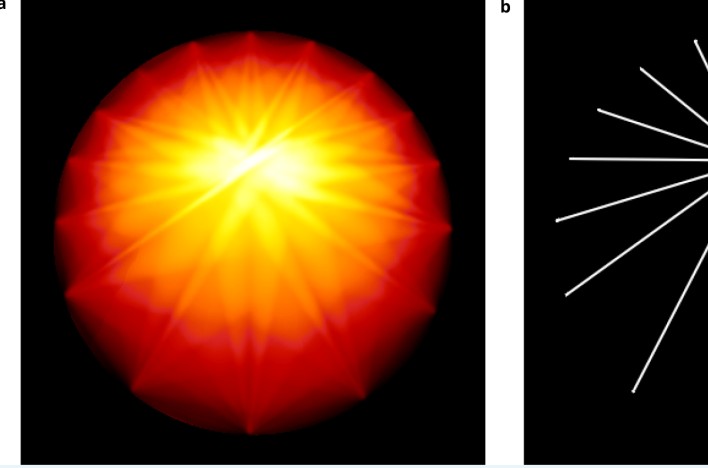

**Appendix 6—figure 1.** Uniformity analysis. (**a**) Entropy map obtained on a simulated image sequence (see *Appendix 1—figure 2d*) showing at each point the angular distribution entropy obtained when considering this point as the reference. (**b**) Estimated reference points (green disks) obtained for 10 different simulated image sequences.

DOI: https://doi.org/10.7554/eLife.32311.036

