## [Decision Letter]

Thank you for submitting your article "A quantitative approach for the spatio-temporal distribution of 3D intracellular events in fluorescence microscopy" for consideration by *eLife*. Your article has been reviewed by three peer reviewers, and the evaluation has been overseen by Anna Akhmanova as the Senior/Reviewing Editor. The reviewers have opted to remain anonymous.

The reviewers have discussed the reviews with one another and the Reviewing Editor has drafted this decision to help you prepare a revised submission.

Summary:

The paper presents a software tool (QuantEv) for quantitative analysis and visualization of the spatial distribution of intracellular events imaged by fluorescence microscopy and represented by static or dynamic descriptors associated with spatial coordinates. The potential practical value of the tool is demonstrated by studying the distribution of moving Rab6 fluorescently labeled membranes with respect to their direction of movement in differently shaped cells, as well as the position of the generating hub of Rab11 positive membranes, and the effects of actin disruption on Rab11 trafficking in relation to cell shape. The paper is well written, and the results are interesting.

Essential revisions:

1) The manuscript ignores extensive prior relevant work on similar problems. The analysis methods described are simple, and not particularly innovative. Evidence of generalizability is lacking. By not considering and incorporating prior approaches, the potential capabilities and applicability of the software have been significantly limited. It is especially important in describing new software to compare its performance to existing approaches. These comparisons should include not just one similar method (kernel density estimation) that is not widely used but other methods (e.g., http://doi.org/10.1038/nmeth.1486). For example, although developed for a different application, the software tool plusTipTracker (http://dx.doi.org/10.1016/j.jsb.2011.07.009) can also perform various dynamics analyses related to cell location, and should be discussed. The same goes for TrackMate (http://doi.org/10.1016/j.ymeth.2016.09.016). More generally, some discussion is needed of what, exactly, can and cannot be done with existing tools, to make the novelties and benefits of the proposed tool more explicit.

2) It seems that the statistical test the authors are proposing is focusing on comparing two groups using the Wilcoxon signed-rank test. While this is fine as such, often in Biology more than two groups need to be compared with each other; like wt, mutant, and rescue for example. Also, the possibility to compare more than two groups is important to avoid problems with repetitive testing between groups. The problem is the additive per comparison error. This issue should be addressed, and potential solutions should be included in the next version.

3) The authors suggest using intensity as a weight for the analysis. In the cases used in the synthetic test images test and likely in the experimental data here the intensities are comparable. However, the potential user needs to be instructed that appropriate normalization procedures need to be applied in case that intensities are used as weight. Likewise, the segmentation needs to start from a similar selection. The authors should at least discuss this necessity and provide what the prerequisites are for the input.

4) The only datasets used in paper are artificial, in that cells were constrained to specific geometries. This reduces the inherent complexity with unknown other effects. Most investigators would not choose to use artificial geometric constraints, and no analysis is presented for images of cells that show natural variation in shape, either in vitro or in vivo. Such variation might overwhelm the straightforward approaches the authors describe and this possibility should be investigated. Application of the methods to an image dataset for unconstrained cells should be included.

5) The claim that the proposed framework is "generic and non-parametric" seems too strong. In the paper only a few very specific applications are investigated. And many of the underlying components of the framework are not exactly non-parametric. For example, the kernels involved in the weighted density estimation have parameters, and the distance measures depend on the number of bins. This claim should be toned down.

6) The authors claim that their framework is more sensitive than the Kernel density maps. This asks the question of the discriminatory power of the method. The potential to differentiate distribution patterns depends on the resolution of the input; this should be discussed.

7) The meaning of the results of the various analyses is often unclear and very limited in terms of providing understanding of mechanisms for any of the systems studied. Since the paper is written for a general audience, this should be improved.

8) A number of specific comments on the text must be addressed:

Introduction, first paragraph “Automatic methods have the obvious advantage of being quicker and reproducible. However, most computational methods are based on the complex combination of heterogeneous features such as statistical, geometrical, morphological and frequency properties (Peng, 2008), whichmakes difficult to draw de1nitive biological conclusions”: This statement ignores extensive work on generative or mechanistic models, which produce interpretable parameters. Such work includes mechanistic models of dynamics of endocytic vesicles (e.g., http://doi.org/10.1038/nmeth.1237) and cytoskeletal dynamics (http://doi.org/10.1126/science.1100533), and generative models of vesicle distribution (e.g., http://doi.org/10.1371/journal.pcbi.1004614).

Introduction, first paragraph “Additionally, most experimental designs, especially at single cell level, pool together data coming from replicated experiments of a given condition (Schauer et al., 2010; Merouane et al., 2015; Biot et al., 2016), neglecting the biological variability between individual cells.”: Again, this ignores work on generative models that specifically analyzes and captures variation between cells. Past examples include microtubule networks (e.g., http://doi.org/10.1371/journal.pone.0050292), and cell and nuclear shape (e.g., http://dx.doi.org/10.1091/mbc.E15-06-0370). Traditional feature-based methods also frequently analyze heterogeneity within populations (e.g., http://doi.org/10.1371/journal.pone.0102678).

Introduction, third paragraph and subsection “Weighted density estimation”, first paragraph – The use of circular and/or cylindrical coordinate systems for description of object positions within a cell is well established (e.g., http://doi.org/10.1002/cyto.a.20487 and http://doi.org/10.1002/cyto.a.21066) and in these cases rotation angle was more powerfully defined relative to the major axis of each cell rather than being defined by the confinement fields. Alternative approaches to the problem, such as morphing, were not discussed.

Introduction, third paragraph and subsection “Distance between densities”, first paragraph – There is no discussion of more recent metrics related to Earth Mover's Distance that have been described and used to compare subcellular patterns (http://doi.org/10.1007/s11263-012-0566-z).

"The KD approach concludes… Instead, QuantEv selectively identifies…" How do we know which method comes closest to the truth? Can this be verified? Without some control experiment or simulation, how can we conclude that QuantEv is to be preferred over other methods?

Subsection “Visualizing and quantifying the influence of micropatterns on the spatial distribution of Rab6 positive membranes” – There is no clear peak at the two-thirds position in Figure 2D and no evidence of significance or reproducibility is presented.

"Rab6 trajectories were classified into two categories…" How do we know these trajectories are trustworthy? What kind of control experiment was performed to confirm this? This is especially important since it seems the trajectories were not obtained with the best methods available these days (for example according to http://doi.org/10.1038/nmeth.2808 there seem to be better tracking methods than the method mentioned in the subsection “Event detection and localization”).

"we extracted Rab11 trajectories…" Same as previous comment.

"On the image sequences considered in the previous section (see Figure 4A), this distance remains stable (see Figure 5A). We analyzed cells treated with Latrunculin A… the ERC location is moving away as the drug is affecting the cell (see Figure 5B)". But the time scales are very different in these two cases (seconds versus minutes). Control experiments would be needed to confirm that the ERC location in non-treated cells remains stable over the same time scale as in the treated cells.

The claim that the presented software tool "is efficient with small and large amounts of data" has not been demonstrated in the paper. Neither dataset sizes nor processing times are mentioned.

The claim that "QuantEv is quite flexible since the user can specify any distance…" contradicts the statement that "it is fully automated and non-parametric".

"a reference point… and a reference direction have to be specified by the user…" Same as previous comment.

[Editors' note: further revisions were requested prior to acceptance, as described below.]

Thank you for resubmitting your work entitled "A quantitative approach for the spatio-temporal distribution of 3D intracellular events in fluorescence microscopy" for further consideration at *eLife*. Your revised article has been favorably evaluated by Anna Akhmanova (Senior/Reviewing Editor) and three reviewers.

The manuscript has been improved but there are some remaining issues that need to be addressed before acceptance, as outlined below:

There were significant reservations about using constrained cells, as the use of such cells greatly simplifies the analysis. The authors have now done additional work to add results comparing unconstrained cells and two different types of constrained cells. The results show that QuantEV is able to distinguish among the three groups. However, no perturbation studies (e.g., Latrunculin B) were done with unconstrained cells. Thus the main concern remains about the suitability of QuantEV for use in future studies, the majority of which are expected to be done with unconstrained cells. This is an important point: the method may be able to distinguish changes within constrained cells upon various treatments, but may not be able to distinguish perturbations on the background of significant variation within unconstrained cells. There is no information provided on the variance of the profiles in Figure 2F within the unconstrained population. This is a major concern in the context of the very broad claims made in the manuscript (especially in the Discussion) about the power and generality of QuantEV. To support these broad claims, the authors must provide conclusive evidence that QuantEV can distinguish physiologically relevant changes upon perturbations in unconstrained cells. Since the necessary datasets are undoubtedly available to the authors, no collection of new experimental data is expected to be necessary to address this point.

---

## [Author Response]

Essential revisions:1) The manuscript ignores extensive prior relevant work on similar problems. The analysis methods described are simple, and not particularly innovative. Evidence of generalizability is lacking. By not considering and incorporating prior approaches, the potential capabilities and applicability of the software have been significantly limited. It is especially important in describing new software to compare its performance to existing approaches. These comparisons should include not just one similar method (kernel density estimation) that is not widely used but other methods (e.g., http://doi.org/10.1038/nmeth.1486). For example, although developed for a different application, the software tool plusTipTracker (http://dx.doi.org/10.1016/j.jsb.2011.07.009) can also perform various dynamics analyses related to cell location, and should be discussed. The same goes for TrackMate (http://doi.org/10.1016/j.ymeth.2016.09.016). More generally, some discussion is needed of what, exactly, can and cannot be done with existing tools, to make the novelties and benefits of the proposed tool more explicit.

We agree that plusTipTracker and TrackMate showed analyses that associate dynamic features with localization, and we failed to report these studies in the first version of the manuscript. However, plusTipTracker and TrackMate do not provide a framework to quantitatively analyze the dynamic features with respect to intracellular localization, which QuantEv does. Actually, neither precise way to look at localization nor statistical tool are available with these tracking methods. Nonetheless, studying dynamic features such as confinement ratio or lifetime with respect to their intracellular localization is not currently available for biologists while it potentially represents a routine analysis for tracking experiments. Consequently, we proposed to the developers of TrackMate to add a QuantEv analysis module in their plugin and to the Icy developers to add a QuantEv track processor. They both accepted, demonstrating a need for this type of analysis. We are currently collaborating with Jean-Yves Tinevez, TrackMate developer, to implement a QuantEv analysis module (https://github.com/tpecot/QuantEvForTrackMate), the QuantEv track processor will follow. We added a new section explaining how we compute histograms and densities of dynamical features with respect to localization (subsection “Density estimation for dynamical features”). We also changed the second paragraph of the Results to analyze the radial distribution of confinement ratio, total path length and lifetime instead of using these features as weights in the densities.

2) It seems that the statistical test the authors are proposing is focusing on comparing two groups using the Wilcoxon signed-rank test. While this is fine as such, often in Biology more than two groups need to be compared with each other; like wt, mutant, and rescue for example. Also, the possibility to compare more than two groups is important to avoid problems with repetitive testing between groups. The problem is the additive per comparison error. This issue should be addressed, and potential solutions should be included in the next version.

We thank the reviewers for this remark, we actually did not think about comparing more than two groups together. The statistical test is applied on the difference between the average inter-distance and the average intra-distance for each image sequence in the study. Considering more than two groups does not change the average intra-distance and only expands the average inter-distance to more than one group. Accordingly, we changed the text in the section entitled “Difference between conditions” and modified the plugin to compare more than two groups.

3) The authors suggest using intensity as a weight for the analysis. In the cases used in the synthetic test images test and likely in the experimental data here the intensities are comparable. However, the potential user needs to be instructed that appropriate normalization procedures need to be applied in case that intensities are used as weight. Likewise, the segmentation needs to start from a similar selection. The authors should at least discuss this necessity and provide what the prerequisites are for the input.

We acknowledge that intensity normalization is an important step in the analysis of fluorescent images and needs to be discussed. As intensity is proportional to the amount of proteins in fluorescence microscopy, it potentially provides useful information. However, several phenomena such as photobleaching, phototoxicity, shading or uneven illumination potentially alter this proportionality. If the user is able to correct for these phenomena, it is preferable to use intensity as weights for the analysis. Otherwise, it is safer not to use it. We added a paragraph in the Discussion (third paragraph) to address this point.

4) The only datasets used in paper are artificial, in that cells were constrained to specific geometries. This reduces the inherent complexity with unknown other effects. Most investigators would not choose to use artificial geometric constraints, and no analysis is presented for images of cells that show natural variation in shape, either in vitro or in vivo. Such variation might overwhelm the straightforward approaches the authors describe and this possibility should be investigated. Application of the methods to an image dataset for unconstrained cells should be included.

Thank you for this valuable suggestion. In the revised manuscript, we added a set of image sequences with Rab6 positive membranes in unconstrained cells (Appendix 1 Figure 1A) and compared them with crossbow- and disk-shaped cells (subsections “Visualizing and quantifying the influence of cell shape on the spatial distribution of Rab6 positive membranes” and “Inwards and outwards Rab6 positive membranes show two distinctive dynamical behaviors”, Figures 2-4). Additionally, these sequences were acquired with a different modality, demonstrating that QuantEv allows us to compare image sequences with different cell shapes and acquired with different modalities. This study demonstrates that QuantEv is suited to compare images coming from different databases and laboratories, acquired recently or several years ago.

5) The claim that the proposed framework is "generic and non-parametric" seems too strong. In the paper only a few very specific applications are investigated. And many of the underlying components of the framework are not exactly non-parametric. For example, the kernels involved in the weighted density estimation have parameters, and the distance measures depend on the number of bins. This claim should be toned down.

As described in the section “Weighted density estimation”, the bandwidths of the Gaussian kernels are estimated with the Silverman’s rule of thumb and the concentrations of the von Mises kernels are estimated with the robust rule of thumb proposed by Taylor et al. These usual parameters in non-parametric density estimation are consequently automatically estimated and the user does not have to set them. Meanwhile, as the statistical test used in the QuantEv framework is based on the ranks of a difference between earth mover’s distances or circular earth mover’s distances, the result of the test is not affected by the choice of the number of bins. In the Appendix, we evaluated the influence of binning on the earth mover’s distance (see Appendix 4) to clarify this point. Actually, the user just needs to provide the coordinate system, the coordinate system center and a reference direction. For all these reasons, QuantEv can be considered as “semi-parametric framework” for traffic phenotype analysis. We added a full paragraph in the Discussion about this point to clearly state what are the needed inputs by the user (second paragraph).

6) The authors claim that their framework is more sensitive than the Kernel density maps. This asks the question of the discriminatory power of the method. The potential to differentiate distribution patterns depends on the resolution of the input; this should be discussed.

QuantEv is able to significantly detect differences if the phenotypes are positively different. The Kernel Density maps approach is too sensitive as it wrongly detects differences for image sequences showing Rab6 proteins trafficking in cells with the same shape (see Figure 2D). The usual ANOVA test also fails as shown in simulations in Appendix 3. In this experiment, the ANOVA test leads to statistical significance when comparing image sequences with different distributions but also when comparing image sequences with same or mixed distributions when the amount of data becomes large (see Appendix 3). The same experiment demonstrates that QuantEv accurately identifies differences for sequences with different distributions with a small number of image sequences, but does not lead to significant differences for sequences with same or mixed distributions, even with a large number of image sequences, demonstrating a good discriminating power.

7) The meaning of the results of the various analyses is often unclear and very limited in terms of providing understanding of mechanisms for any of the systems studied. Since the paper is written for a general audience, this should be improved.

Maybe we misunderstood the comment. Our intention was not to propose a method for deciphering mechanisms related to Rab6 and Rab11. Actually, these proteins are known to be involved in dedicated molecular complexes and interact with other molecules (e.g. Rab11 interact with actin via Myosin VB et Rab11FIP2). The underlying mechanisms of Rab proteins can be better elucidated if two or more fluorescent markers are used. Probably, a generative model would be helpful to analyze the mechanisms. Instead we propose here a computational framework to compare traffic phenotypes related to cell shape, cytoskeleton organization and spatial organization of organelles.

8) A number of specific comments on the text must be addressed:Introduction, first paragraph “Automatic methods have the obvious advantage of being quicker and reproducible. However, most computational methods are based on the complex combination of heterogeneous features such as statistical, geometrical, morphological and frequency properties (Peng, 2008), whichmakes difficult to draw de1nitive biological conclusions”: This statement ignores extensive work on generative or mechanistic models, which produce interpretable parameters. Such work includes mechanistic models of dynamics of endocytic vesicles (e.g., http://doi.org/10.1038/nmeth.1237) and cytoskeletal dynamics (http://doi.org/10.1126/science.1100533), and generative models of vesicle distribution (e.g., http://doi.org/10.1371/journal.pcbi.1004614).Introduction, first paragraph “Additionally, most experimental designs, especially at single cell level, pool together data coming from replicated experiments of a given condition (Schauer et al., 2010; Merouane et al., 2015; Biot et al., 2016), neglecting the biological variability between individual cells.”: Again, this ignores work on generative models that specifically analyzes and captures variation between cells. Past examples include microtubule networks (e.g., http://doi.org/10.1371/journal.pone.0050292), and cell and nuclear shape (e.g., http://dx.doi.org/10.1091/mbc.E15-06-0370). Traditional feature-based methods also frequently analyze heterogeneity within populations (e.g., http://doi.org/10.1371/journal.pone.0102678).

We improved the state-of-the-art and we referenced the aforementioned modeling approaches. These methods are generative or dedicated methods to analyze specific dynamics. QuantEv is more generic and enables to analyze the spatial distribution of intracellular events with no prior on dynamics. It can be understood as a statistical tool to detect significance evidence between phenotypes for a large range of applications. We added a paragraph in the Discussion (third paragraph) to guide the user about when to use QuantEv and when to use generative or mechanistic models.

Introduction, third paragraph and subsection “Weighted density estimation”, first paragraph – The use of circular and/or cylindrical coordinate systems for description of object positions within a cell is well established (e.g., http://doi.org/10.1002/cyto.a.20487 and http://doi.org/10.1002/cyto.a.21066) and in these cases rotation angle was more powerfully defined relative to the major axis of each cell rather than being defined by the confinement fields. Alternative approaches to the problem, such as morphing, were not discussed.

We included the aforementioned references as suggested (Appendix 2). Actually, spherical and cylindrical representations have already been considered in many approaches. The added value of QuantEv is mainly to analyze spatial and temporal information in a common and appropriate reference system. Moreover, our aim was no to create an atlas of traffic based on a set of points. Morphing registration is probably the best approach to align cells in a common reference but it is generally based on shape features and contours. In our study, we evaluate several vesicle trafficking and the number of detected events is different in each cell. We normalized the distances to avoid morphing cells of different shapes.

Introduction, third paragraph and subsection “Distance between densities”, first paragraph – There is no discussion of more recent metrics related to Earth Mover's Distance that have been described and used to compare subcellular patterns (http://doi.org/10.1007/s11263-012-0566-z).

Thank you, we added this reference to the first paragraph of the subsection “Distance between densities” with the reference to https://doi.org/10.1073/pnas.1319779111.

"The KD approach concludes… Instead, QuantEv selectively identifies…" How do we know which method comes closest to the truth? Can this be verified? Without some control experiment or simulation, how can we conclude that QuantEv is to be preferred over other methods?

We acknowledge that this point was not clear enough in the previous version of the manuscript. In this new version of the manuscript, we emphasized the fact that the KD approach leads to statistically significant results when comparing image sequences with cells of same shape (subsection “Visualizing and quantifying the influence of cell shape on the spatial distribution of Rab6 positive membranes”, Figure 2D), demonstrating the KD approach is too sensitive, which is not the case with QuantEv.

Subsection “Visualizing and quantifying the influence of micropatterns on the spatial distribution of Rab6 positive membranes” – There is no clear peak at the two-thirds position in Figure 2D and no evidence of significance or reproducibility is presented.

We acknowledge that the term “peak” might be misleading. We replaced it with maxima to avoid confusion.

"Rab6 trajectories were classified into two categories…" How do we know these trajectories are trustworthy? What kind of control experiment was performed to confirm this? This is especially important since it seems the trajectories were not obtained with the best methods available these days (for example according to http://doi.org/10.1038/nmeth.2808 there seem to be better tracking methods than the method mentioned in the subsection “Event detection and localization”)."we extracted Rab11 trajectories…" Same as previous comment.

We agree that only relying on one tracking method is risky. We also processed the image sequences with the methods proposed by Sbalzarini and Koutmoutsakos and TrackMate, so we now have a multiple hypothesis tracking method, a combinatorial optimization tracking method and a hybrid approach. We then computed a gated distance as defined in Chenouard et al. Between all the trajectories estimated with the three methods, we selected the trajectories for which this distance was inferior to 2 pixels in at least two methods for the analysis (subsection “Event detection and localization”). These selected tracks were used to draw conclusions and detect evidence about phenotypes.

"On the image sequences considered in the previous section (see Figure 4A), this distance remains stable (see Figure 5A). We analyzed cells treated with Latrunculin A… the ERC location is moving away as the drug is affecting the cell (see Figure 5B)". But the time scales are very different in these two cases (seconds versus minutes). Control experiments would be needed to confirm that the ERC location in non-treated cells remains stable over the same time scale as in the treated cells.

Thank you for this valuable remark. We agree and included new experiments with disk-shaped cells without latrunculin A treatment acquired every 30 seconds for 20 minutes. Figure 6 was changed accordingly.

The claim that the presented software tool "is efficient with small and large amounts of data" has not been demonstrated in the paper. Neither dataset sizes nor processing times are mentioned.

In Appendix 3, we show how large amount of data can be a problem when using a regular ANOVA analysis. This is not the case with QuantEv that accurately identifies simulations with different distributions but does not state any statistical difference between sequences with same or mixed distributions, even with a large number of image sequences. We also added the Appendix 5 to report the processing times for the different analyzes proposed in QuantEv.

The claim that "QuantEv is quite flexible since the user can specify any distance…" contradicts the statement that "it is fully automated and non-parametric"."a reference point… and a reference direction have to be specified by the user…" Same as previous comment.

We refer the reviewers to our answer to the point 5 raised in the main essential revision.

[Editors' note: further revisions were requested prior to acceptance, as described below.]

The manuscript has been improved but there are some remaining issues that need to be addressed before acceptance, as outlined below:There were significant reservations about using constrained cells, as the use of such cells greatly simplifies the analysis. The authors have now done additional work to add results comparing unconstrained cells and two different types of constrained cells. The results show that QuantEV is able to distinguish among the three groups. However, no perturbation studies (e.g., Latrunculin B) were done with unconstrained cells. Thus the main concern remains about the suitability of QuantEV for use in future studies, the majority of which are expected to be done with unconstrained cells. This is an important point: the method may be able to distinguish changes within constrained cells upon various treatments, but may not be able to distinguish perturbations on the background of significant variation within unconstrained cells. There is no information provided on the variance of the profiles in Figure 2F within the unconstrained population. This is a major concern in the context of the very broad claims made in the manuscript (especially in the Discussion) about the power and generality of QuantEV. To support these broad claims, the authors must provide conclusive evidence that QuantEV can distinguish physiologically relevant changes upon perturbations in unconstrained cells. Since the necessary datasets are undoubtedly available to the authors, no collection of new experimental data is expected to be necessary to address this point.

To demonstrate that QuantEv is able to distinguish changes induced by perturbation studies, we have run a new series of experiments consisting of the acquisition of Rab11 positive membranes in unconstrained cells after Latrunculin A treatment. As unconstrained cells are more spread out and have a less organized cytoskeleton than cells on micro-patterns, they show a weaker Latrunculin A resistance. Consequently, unconstrained RPE1 cells shrink faster than constrained cells, leading to a complete detachment from the slide approximately 20 minutes after Latrunculin A injection. This does not happen for constrained RPE1 cells as they are more stably attached to the slide through the micro-patterns. In order to compare constrained and unconstrained cells, we only kept image sequences at 10 and 15 minutes after Latrunculin A treatment, removing from the previous version of the manuscript the image sequences acquired at 20 and 25 minutes after Latrunculin A treatment for constrained cells. It has to be noted that the radial distributions observed at 20 and 25 minutes after Latrunculin A treatment for constrained cells are very similar to the radial distributions observed at 15 minutes after treatment. The Latrunculin A effect on the dynamical behavior of Rab11 positive membranes is similar in unconstrained and constrained cells (see Figure 7). Furthermore, Latrunculin A injection induces a shift of the radial distribution of Rab11 positive membranes from the cell periphery to the cell center in unconstrained cells, a phenomenon also observed in constrained cells (see Figure 8A). Finally, the difference between radial distributions of Rab11 positive membranes in the three different conditions (unconstrained, crossbow- and disk-shaped cells) at Latrunculin A injection time is statistically significant (see Figure 8B) while the differences of the same radial distributions 10 and 15 minutes after Latrunculin A injection are not (see Figure 8B). This emphasizes that Latrunculin A influence on radial distribution is similar for the three conditions. We modified accordingly the subsection entitled “Joint actin disruption and cell shape influence on Rab11 radial distribution”. We believe that the addition of a perturbation study in unconstrained cells supports the power and generality of QuantEv and we thank the reviewers for this valuable suggestion. There is no information provided on the variance of the profiles in Figure 2F within the unconstrained population. This is a major concern in the context of the very broad claims made in the manuscript (especially in the Discussion) about the power and generality of QuantEV. The variance of the profiles are not shown on the graphs as it is difficult to add other curves on such packed graphs. However, the fact that the statistical test applied to the radial distributions is low (pvalue = 7.3x10^-4^) demonstrates that the variance of the profiles has to be small. Actually, the averaged standard deviation per bin for the radial distribution of Rab6 positive membranes in unconstrained cells is equal to 0.0074. For each analysis in the manuscript, a statistical test is performed to both demonstrate differences between distributions and take into account the variability in each dataset.